# Dense associative memory for Gaussian distributions

**Chandan Tankala** [1]  **Krishnakumar Balasubramanian** [2]

## Abstract

Dense associative memories (DAMs) store and retrieve patterns via energy-function based fixed points, but existing models are limited to vector representations. We extend DAMs to Gaussian densities equipped with the 2-Wasserstein distance. Our framework defines a log-sum-exp energy over stored distributions and a retrieval dynamics aggregating optimal transport maps in a Gibbs-weighted manner. Stationary points correspond to self-consistent Wasserstein barycenters, generalizing classical DAM fixed points. We prove exponential storage capacity and provide quantitative retrieval guarantees under Wasserstein perturbations. We validate the method on synthetic and real-world image (CelebA and CIFAR-10 datasets) and text (text8 and NLI corpus) datasets. By generalizing from vectors to distributions, our work bridges classical DAMs with modern generative modeling and paves the way for distributional storage and retrieval in memory-augmented learning.

## 1. Introduction

Associative memories are a foundational paradigm for robust storage and retrieval of structured information. Classical models, such as the Hopfield network (Hopfield, 1982a) and its modern high-capacity extensions (Krotov & Hopfield, 2016; Ramsauer et al., 2020), demonstrate that high-dimensional patterns can be stored in distributed representations and retrieved accurately under partial or corrupted queries. These models formalize the principle that memory retrieval can be framed as a dynamical system evolving toward energy minima corresponding to stored patterns.

While prior work on associative memory has focused on

vector-valued data, many modern applications involve *probability distributions* as fundamental objects. Gaussian embeddings, in particular, have become a versatile framework for capturing uncertainty: words (Vilnis & McCallum, 2014), graph nodes (Bojchevski & Günnemann, 2018), documents (Gourru et al., 2021), and sentences (Yoda et al., 2024) have all been embedded as Gaussian distributions. These Gaussian representations naturally admit similarity measures via Wasserstein distance, yet existing associative memory frameworks cannot operate on them directly. More recently, results on Joint-Embedding Predictive Architectures (JEPAs) show that the anti-collapse (diversity) regularization term in JEPA explicitly drives a self-supervised encoder to produce Gaussian embeddings, implying that the encoder necessarily internalizes the data density (Balestriero et al., 2025; Zimmermann et al., 2025). This provides further evidence suggesting that Gaussian embeddings arise naturally in modern representation learning and motivates the need for memory architectures that can operate directly on distributional representations.

More generally, uncertainty-aware generative modeling relies on probabilistic models such as variational autoencoders (Kingma & Welling, 2013), normalizing flows (Rezende & Mohamed, 2015), and diffusion models (Sohl-Dickstein et al., 2015; Ho et al., 2020) that learn distributions over complex modalities including images, text, and 3D point-clouds. Similarly, in Bayesian inference, posterior beliefs about latent variables are encoded as probability densities (often Gaussian or Gaussian mixtures) where updating or recalling these beliefs corresponds to operations directly on distributions (Bernardo & Smith, 2009; Khan & Rue, 2023). In these contexts, it is natural to treat entire distributions, rather than individual samples, as primary computational units.

The aforementioned considerations motivate the following question:

> *Can associative memories be generalized to store and retrieve probability distributions, rather than deterministic vectors?*

We address this question in the setting of Gaussian distributions. Let $\mathcal{N}(\mu_i, \Sigma_i)$, $i = 1, \ldots, N$, be the target distributions. Endowed with the *Bures–Wasserstein geometry*

---

[1]Independent Researcher [2]Department of Statistics, University of California, Davis, USA. Correspondence to: Chandan Tankala <chandanusair87@gmail.com>, Krishnakumar Balasubramanian <kbala@ucdavis.edu>.

*Proceedings of the 43$^{rd}$ International Conference on Machine Learning*, Seoul, South Korea. PMLR 306, 2026. Copyright 2026 by the author(s).

(Asuka, 2011; Lambert et al., 2022; Diao et al., 2023), Gaussians inherit a Riemannian structure from the Wasserstein-2 distance that captures both mean and covariance, providing a natural notion of similarity. Our goal is to design a *dense associative memory (DAM)* that robustly stores $\{\mathcal{N}(\mu_i, \Sigma_i)\}_{i=1}^{N}$ and retrieves the correct distribution from noisy or partial queries, thus extending classical DAMs from $\mathbb{R}^d$ to the non-Euclidean space of probability measures while retaining high capacity and robust retrieval. Toward that end, we make the following **contributions** in this work:

1. **Wasserstein LSE energy functional.** We propose a novel energy formulation defined directly on the Bures-Wasserstein space, generalizing classical DAM energies to probability densities (see Section 2). We provide detailed theoretical comparisons between Bures-Wasserstein DAM and a natural baseline–performing Euclidean DAM on the mean and covariance matrix stacked as a vector–highlighting differences in separation conditions, fixed-point analysis, and basin geometry (see Remarks 1 and 2).

2. **Exponential storage capacity.** We prove that our model achieves storage capacity exponential in the dimensionality of the ambient space, extending classical vectorial results to the Gaussian distribution setting (see Theorem 1).

3. **Retrieval guarantees.** We establish bounds on the fidelity of retrieval under noisy query distributions, providing explicit dependence on the Wasserstein distance between stored and perturbed densities (see Section 3.2).

4. **Empirical validation.** Through experiments on synthetic Gaussian datasets and real-data experiments on Gaussian word, sentence, and image Gaussian embeddings applied to standard benchmark datasets, we demonstrate that the proposed model achieves accurate retrieval and exhibits the robustness predicted by our theoretical analysis. (see Section 4)

By extending dense associative memories from vectors to probability densities, our work lays a foundation for *distributional memory architectures* in generative AI. Such memories can store, recall, and manipulate probabilistic objects, enabling memory-augmented probabilistic reasoning and uncertainty-aware generative computation.

**Notation and definitions.**

*Basic notation.* For a positive integer $N$, define $[N] := \{1, 2, \ldots, N\}$. We write $\|\cdot\|$ for the Euclidean norm and $\langle \cdot, \cdot \rangle_{L^2}$ for the $L^2$ inner product between probability measures. Throughout, we work in the space $\mathcal{P}_2(\mathbb{R}^d)$ of probability measures with finite second moment, equipped with

the 2-Wasserstein distance

$$W_2(\mu, \nu) = \inf_{\gamma \in \Gamma(\mu, \nu)} \left( \int_{\mathbb{R}^d \times \mathbb{R}^d} \|x - y\|^2 \, d\gamma(x, y) \right)^{1/2},$$

where $\Gamma(\mu, \nu)$ is the set of couplings with marginals $\mu, \nu$. For a measurable $T : X \to Y$ and $\mu$ on $X$, the push-forward is $T_{\#}\mu(B) := \mu(T^{-1}(B))$ for measurable $B \subseteq Y$.

*Variational structure.* For a functional $\mathcal{F} : \mathcal{P}_2(\mathbb{R}^d) \to \mathbb{R}$, its first variation at measure $\mu$ is the function $\frac{\delta \mathcal{F}}{\delta \mu}(\mu)(x)$ defined by

$$\frac{d}{d\varepsilon} \mathcal{F}(\mu + \varepsilon(\mu' - \mu))\big|_{\varepsilon=0} = \int \frac{\delta \mathcal{F}}{\delta \mu}(\mu)(x) \, (\mu' - \mu)(x) \, dx.$$

The Wasserstein gradient is defined as

$$\nabla_{\mathcal{W}} \mathcal{F}(\mu)(x) := \nabla_x \frac{\delta \mathcal{F}}{\delta \mu}(\mu)(x),$$

and the associated (negative) gradient flow is the continuity equation

$$\partial_t \mu_t + \nabla \cdot (\mu_t v_t) = 0, \quad \text{where } v_t(x) = -\nabla_{\mathcal{W}} \mathcal{F}(\mu_t)(x),$$

often written compactly as $\dot{\mu}_t = -\nabla_{\mathcal{W}} \mathcal{F}(\mu_t)$.

*Gaussian specialization.* The squared 2-Wasserstein distance between two Gaussians admits the closed form

$$W_2^2(\mathcal{N}(\mu_1, \Sigma_1), \mathcal{N}(\mu_2, \Sigma_2)) = \|\mu_1 - \mu_2\|^2 + \text{tr}\left(\Sigma_1 + \Sigma_2 - 2(\Sigma_1^{1/2} \Sigma_2 \Sigma_1^{1/2})^{1/2}\right),$$

where $\text{tr}(\Sigma)$ is the trace of the matrix $\Sigma$. The Bures-Wasserstein gradient at $\mathcal{N}(\mu, \Sigma)$ becomes the projection of the Wasserstein gradient to the tangent space at $\mathcal{N}(\mu, \Sigma)$ (Lambert et al., 2022, page 22) and it further reduces to finite-dimensional gradients:

$$\nabla_{\mathcal{W}} \mathcal{F}(m, \Sigma) = \left( \nabla_m \mathcal{F}(m, \Sigma), \nabla_{\Sigma} \mathcal{F}(m, \Sigma) \right).$$

Moreover, if $X \sim \mathcal{N}(m_0, \Sigma_0)$ and $Y \sim \mathcal{N}(m_1, \Sigma_1)$ with $\Sigma_0, \Sigma_1 \in \mathbb{S}_{++}^d$, the unique optimal transport map $T : \mathbb{R}^d \to \mathbb{R}^d$ pushing $X$ to $Y$ under quadratic cost is affine:

$$T(x) = m_1 + A(x - m_0),$$

where $A = \Sigma_1^{1/2} (\Sigma_1^{1/2} \Sigma_0 \Sigma_1^{1/2})^{-1/2} \Sigma_1^{1/2}$. See Ambrosio et al. (2005); Asuka (2011); Lambert et al. (2022) for additional details.

## 2. Energy Functional in Wasserstein Space

The key design choice in associative memories is the energy function that drives retrieval dynamics. Classical Hopfield networks use a quadratic energy, yielding only $O(d)$ storage capacity in dimension $d$. By contrast, the log-sum-exp

(LSE) energy of Krotov & Hopfield (2016) introduces a sharper nonlinearity and dramatically improves efficiency. For a query $\xi \in \mathbb{R}^d$ and stored patterns $\{X_i\}_{i=1}^N$, the energy is $E(\xi) = -\beta^{-1} \log(\sum_{i=1}^N \exp(-\beta\|X_i - \xi\|^2))$, with temperature parameter $\beta > 0$. As $\beta \to \infty$, this approaches the negative maximum similarity, yielding a smooth approximation to hard maximum retrieval. The induced energy landscape produces well-separated attractor basins, supports exponential storage capacity, and admits a probabilistic view where retrieval corresponds to Gibbs-type aggregation with weights exponentially concentrated on the nearest stored pattern.

A natural first attempt at storing and retrieving Gaussian distributions is to ignore their geometric structure entirely. One can vectorize each Gaussian $\mathcal{N}(\mu, \Sigma)$ by concatenating the mean $\mu \in \mathbb{R}^d$ with the $\frac{d(d+1)}{2}$ upper-triangular entries of $\Sigma$, yielding a representation in $\mathbb{R}^{d+d(d+1)/2}$. Standard Euclidean DAM can then be applied using the update rule of Ramsauer et al. (2020, Equation 3): $\xi_{\text{new}} = \sum_{i=1}^N w_i(\xi)x_i$, where $\{x_i\}_{i=1}^N$ are the vectorized stored patterns, $\xi$ is the vectorized query, and the weights are given by $w_i(\xi) \propto \exp(\beta\langle x_i, \xi\rangle)$. We refer to this baseline as Eu-DAM. However, this approach leads to retrieval failures. Figure 1 illustrates this: we store five well-separated Gaussian measures (see Assumption 1 for the separation condition) in $\mathbb{R}^2$ and initialize queries at Wasserstein distance $0.5r$ from each target, where $r$ is the radius of the contractive ball established in Section 6.3 (see Lemma 4). While our proposed BW-DAM correctly retrieves all five stored Gaussians (Column 2), Eu-DAM fails on two of five retrievals (Column 3), converging instead to incorrect attractors despite identical initialization; see Remarks 1 and 2 for further explanations.

This motivates extending associative memory directly to the space of probability distributions, replacing Euclidean distance with a geometry-preserving similarity measure. Specifically, we work directly in the Wasserstein space $(\mathcal{P}_2(\mathbb{R}^d), W_2)$ and define the LSE energy for stored patterns $X_1, \ldots, X_N \in \mathcal{P}_2(\mathbb{R}^d)$ and query $\xi \in \mathcal{P}_2(\mathbb{R}^d)$ as

$$E(\xi) := -\frac{1}{\beta} \log\left(\sum_{i=1}^N \exp\left(-\beta W_2^2(X_i, \xi)\right)\right). \quad (1)$$

As $\beta \to \infty$, this reduces to the negative minimum Wasserstein distance, implementing a soft-min retrieval rule with a clear probabilistic interpretation: stored distributions are weighted by their Wasserstein proximity to the query. Importantly, this extension preserves the exponential storage capacity of the vector case while operating in a non-Euclidean probability space, making the LSE energy a natural choice for distributional associative memory. Figure 2 shows the LSE energy (1) for five one-dimensional Gaussian measures $\{X_i\}_{i=1}^5$. As $\beta$ increases from 0.1 to 1000, the landscape evolves from nearly flat with overlapping basins to sharp,

well-separated minima. For small $\beta$, discrimination between $\{X_i\}_{i=1}^5$ is weak, while large $\beta$ yields pronounced attractors. Thus, $E$ induces a multi-modal structure in the Bures–Wasserstein geometry, with each $X_i$ serving as an attractor, which is central to our definition of *storage* in Section 3.

The variational structure of our model is encoded through the Wasserstein gradient of the energy functional $E$. By direct differentiation, one obtains $\nabla_{\mathcal{W}} E(\xi) = 2\sum_{i=1}^N w_i(\xi)\,(\mathsf{Id} - T_i)$, where $T_i$ denotes the optimal transport map from $\xi$ to the stored distribution $X_i$, $\mathsf{Id} : \mathbb{R}^d \to \mathbb{R}^d$ is the identity map, and

$$w_i(\xi) := \frac{\exp\left(-\beta W_2^2(X_i, \xi)\right)}{\sum_{j=1}^N \exp(-\beta W_2^2(X_j, \xi))}. \quad (2)$$

The weights $w_i(\xi)$ define a Gibbs-type distribution that assigns higher influence to memories closer to $\xi$ in Wasserstein distance. Thus, the gradient aggregates transport directions from $\xi$ to all stored distributions, with distant contributions exponentially suppressed. The LSE weighting introduces a smooth competitive mechanism, ensuring both robustness of recall and sensitivity to the underlying geometry of the memory ensemble. This formulation directly extends the classical LSE energy functions of dense associative memories from vectors to the Wasserstein space of probability measures.

The retrieval mechanism corresponds to finding stationary points of the energy functional $E$ in the Wasserstein geometry. Setting $\nabla_{\mathcal{W}} E(\xi_*) = 0$ yields $\sum_{i=1}^N w_i(\xi_*)(\mathsf{Id} - T_i) = 0$ or equivalently $\sum_{i=1}^N w_i(\xi_*)T_i = \mathsf{Id}$. In other words, the stationary condition can be written as

$$\left(\sum_{i=1}^N w_i(\xi_*)T_i\right)_\# \xi_* = \xi_*, \quad (3)$$

showing that $\xi_*$ is invariant under the weighted barycentric transport determined by the stored memories. Defining the operator $\Phi : \mathcal{P}_2(\mathbb{R}^d) \to \mathcal{P}_2(\mathbb{R}^d)$ by

$$\Phi(\xi) := \left(\sum_{i=1}^N w_i(\xi)T_i\right)_\# \xi, \quad (4)$$

retrieval is characterized by the fixed points of $\Phi$. In this way, memory recall is expressed as the self-consistency condition $\Phi(\xi_*) = \xi_*$, which generalizes fixed-point equations from classical associative memories to Wasserstein spaces.

## 3. DAM in Bures-Wasserstein Space: Storage and Retrieval

We now specialize our framework to Gaussian distributions, a natural and tractable family for distributional associative

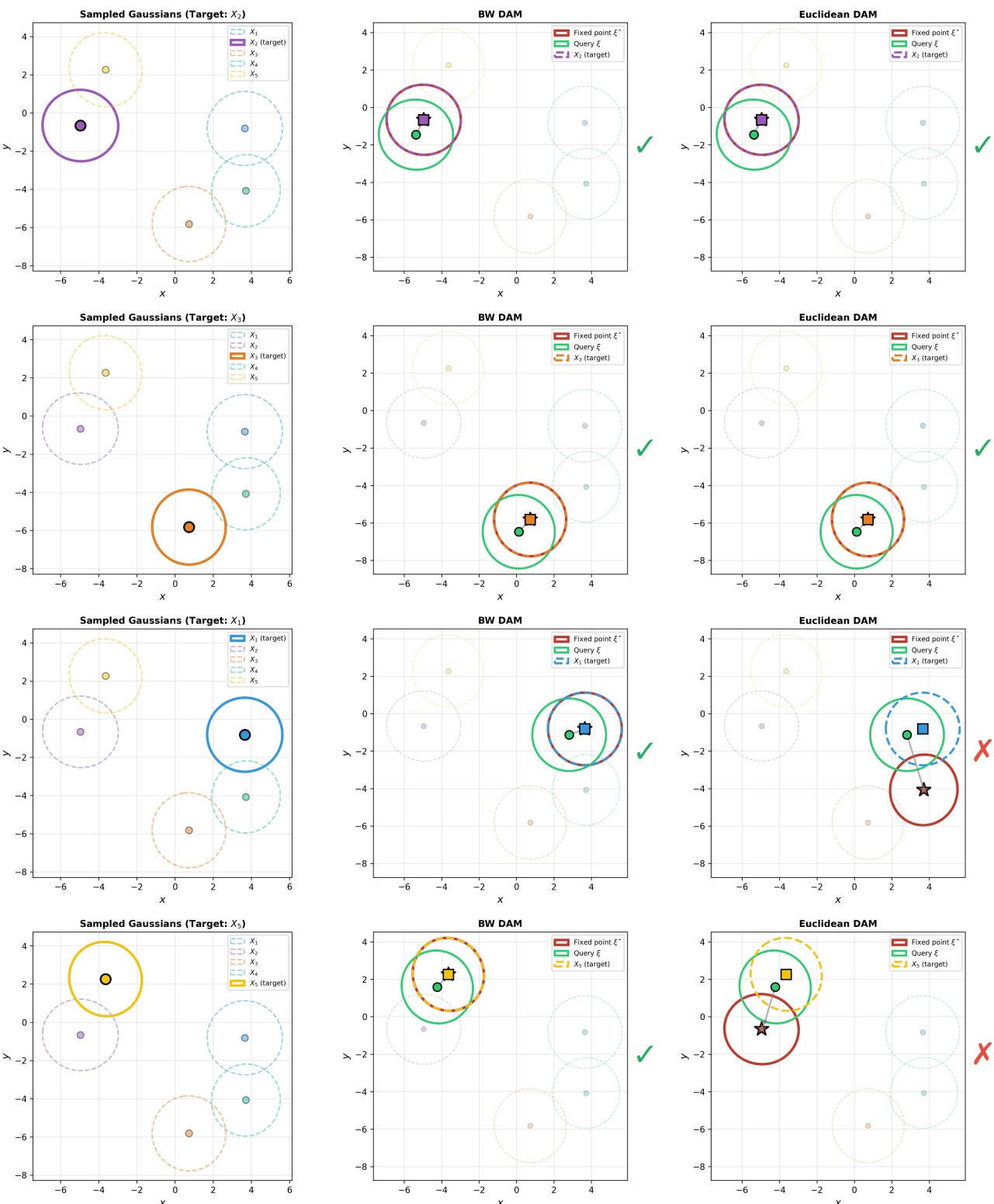

*Figure 1.* Comparison of BW-DAM and Euclidean-DAM dynamics for retrieving stored Gaussian measures. Each row corresponds to perturbing a different stored Gaussian $X_i$ ($i = 1, \ldots, 5$). Column 1: sampled Gaussians with the target highlighted. Columns 2-3: retrieval dynamics showing the query $\xi$ (green), fixed point $\xi^*$ (red), and target (dashed). Parameters: $d = 2$, $N = 5$, $\lambda_{\min} = 0.8$, $\lambda_{\max} = 1.0$, $\beta = 2$, $W_2(\xi, X_i) = 0.5r$ where $r = \sqrt{\lambda_{\min}}$.

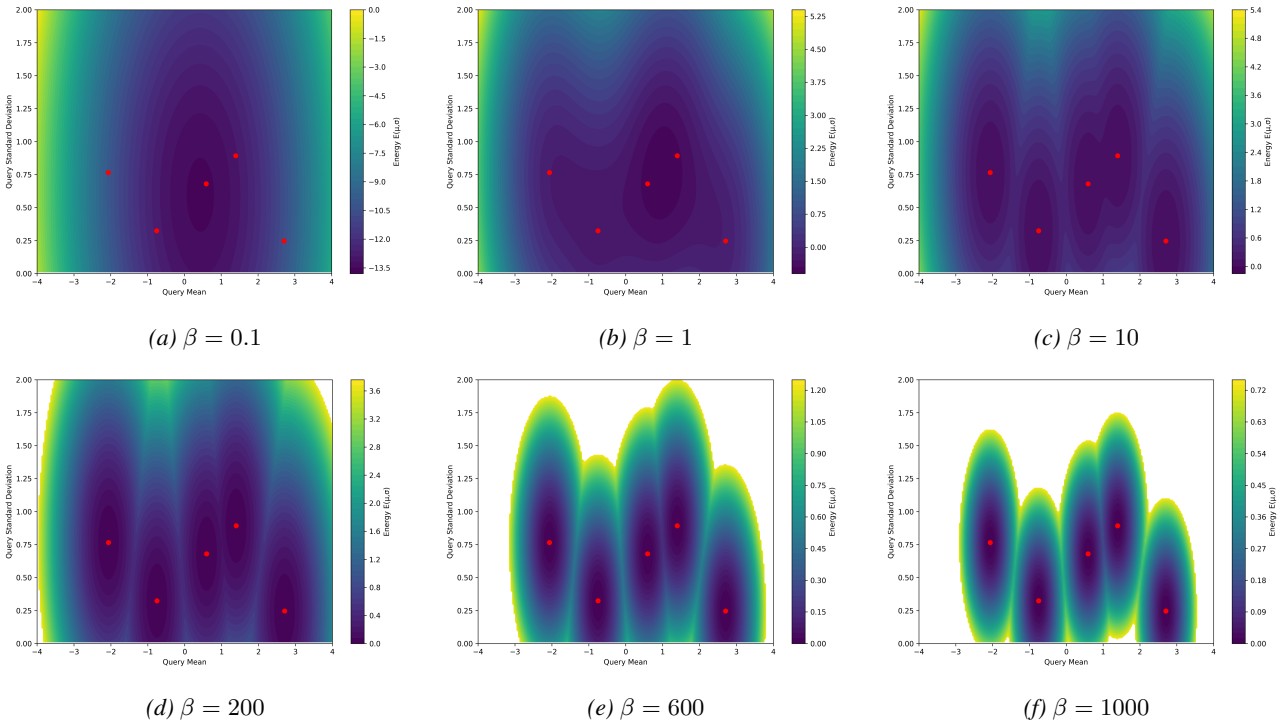

*Figure 2.* Energy landscape $E(\xi)$ (Equation 1) for query one-dimensional Gaussians $\xi = \mathcal{N}(\mu, \sigma^2)$ evaluated on a $200 \times 200$ grid with $\mu \in [-4, 4]$ and $\sigma \in [0.01, 2]$. Red dots indicate $N = 5$ stored Gaussian measures sampled uniformly at random with means in $[-3, 3]$ and standard deviations in $[0.2, 1.0]$. As $\beta$ increases from 0.1 to 1000, the energy transitions from nearly flat with overlapping basins to sharp, well-separated minima.

memories. Gaussians admit a closed-form expression for the 2-Wasserstein distance (Lemma 1), affine optimal transport maps, and broad applicability in machine learning (Section 1). The key challenge is obtaining bounds on errors, which we refer to as the *mean error* and *covariance error* (Lemma 3, Lemma 4). We assume that the eigenvalues of the covariance matrices of stored Gaussian measures lie in the bounded interval $[\lambda_{\min}, \lambda_{\max}]$ with $0 < \lambda_{\min} \leq \lambda_{\max}$.

### 3.1. Storage Capacity

First, we introduce a notion of storage of Gaussian measures in the following definition.

**Definition 1** (Storage of a Gaussian measure). *For a Gaussian measure $X_i$, for $i \in [N]$, consider a Wasserstein ball $\mathcal{B}_i = \{\nu \in \mathcal{P}_2(\mathbb{R}^d) : W_2(X_i, \nu) < r\}$ of radius $r$. The Gaussian measure $X_i$ is defined to be* stored *if there exists a unique fixed point $X_i^* \in \mathcal{B}_i$ to which all $\xi \in \mathcal{B}_i$ converge and $\mathcal{B}_i \cap \mathcal{B}_j = \emptyset$ for all $i \neq j$.*

Next, we prove that exponentially (in dimension $d$) many Gaussian measures can be stored, with high probability, when they are randomly sampled from a Wasserstein sphere.

**Assumption 1** (Separation condition). *Let $X_i = \mathcal{N}(\mu_i, \Sigma_i)$, for $i \in [N]$, be d-dimensional Gaussian measures and suppose the eigenvalues of $\Sigma_i$ lie in the bounded interval*

$[\lambda_{\min}, \lambda_{\max}]$. *Define $\kappa := \lambda_{\max}/\lambda_{\min}$. Suppose $\beta$ is the inverse temperature parameter from the definition of the energy functional in (1). Let $0 < p < 1$ be a constant. Further, define*

$$c := \sqrt{2(2 + \log \kappa)\lambda_{\max}} \ , \quad \alpha := \frac{(1 + \log \kappa)^2}{4(3 + 2\log \kappa)^2} \ ,$$

$$\gamma_1(d) := \left(4c\sqrt{d} + \frac{24\kappa\lambda_{\max}\sqrt{d}}{\sqrt{2\lambda_{\min}}}\right) \times$$
$$\left[4c\sqrt{d} + 4\kappa(3 + \sqrt{d})\left(\sqrt{\lambda_{\min}} + 2c\sqrt{d}\right)\right] \ ,$$

$$\gamma_2(d) := \frac{1}{\sqrt{2\lambda_{\min}}}\left(\frac{48\sqrt{d}\kappa\lambda_{\max}^5}{\lambda_{\min}^{9/2}} + \frac{48\kappa^2\lambda_{\max}}{3\sqrt{\lambda_{\min}}}\right) \ ,$$

$$\beta_0 := \frac{\frac{1}{2}\log p + \alpha d + \log(\gamma_1(d) + \gamma_2(d)) + 2}{2\lambda_{\min}d - 1} \ .$$

*Assume the Gaussian measures $\{X_i\}_{i=1}^N$ satisfy the following separation in $L^2$-inner product: $\min_{i \neq j}(-\log\langle X_i, X_j\rangle_{L^2}) \geq \frac{d}{2}\log(4\pi\lambda_{\max}) + d$.*

**Theorem 1.** *Let $0 < p < 1$ be a constant, and let $\lambda_{\min}, \lambda_{\max}, \kappa, \alpha$ be constants as defined in Assumption 1. Consider a Wasserstein sphere $\mathcal{S}_R$ of radius $R = \sqrt{2d\lambda_{\max}(2 + \log \kappa)}$ centered at $\delta_0$, and let $N = \lfloor\sqrt{p}\, e^{\alpha d}\rfloor$ Gaussian measures be randomly sampled from $\mathcal{S}_R$ using Algorithm 2. Then:*

1. For all $d \geq 4$, the separation condition in Assumption 1 is satisfied with probability at least $1 - p$.

2. There exists $d_0 \in \mathbb{Z}_+$ such that for all $d \geq d_0$ and $\beta \geq \mathsf{max}\{1, \beta_0\}$, where $\beta_0$ is defined in Assumption 1, the energy functional in (1) stores all $N = \lfloor \sqrt{p}\, e^{\alpha d} \rfloor$ randomly sampled Gaussian measures with probability at least $1 - p$.

**Remark 1.** *Theorem 1 extends the exponential storage capacity results of (Ramsauer et al., 2020)[Theorem 3] from Euclidean space to the Bures–Wasserstein space. In the Euclidean setting, separation is measured via inner products $\Delta_i := x_i^\top x_i - \mathsf{max}_{j \neq i} x_i^\top x_j$, and exponential capacity follows from standard concentration arguments on the sphere. In the Bures–Wasserstein setting, we instead measure separation through the $L^2$ inner product between Gaussian measures, which depends non-linearly on means and covariances. To establish high-probability separation, we sample Gaussian measures on a Wasserstein sphere, where the mean directions are uniform on the Euclidean sphere $\mathbb{S}^{d-1}$ and the covariance eigenvalues lie in $[\lambda_{\mathsf{min}}, \lambda_{\mathsf{max}}]$. Concentration of measure on the sphere ensures that the mean directions remain nearly orthogonal across all pairs with high probability, which in turn implies the required $L^2$ separation.*

*However, separation alone does not guarantee storage; we must also show that each stored Gaussian admits a unique fixed point of $\Phi$ in its Wasserstein neighborhood. In Euclidean DAM, the update rule $\xi^{\mathrm{new}} = \sum_{i=1}^{N} w_i(\xi) x_i$ is a weighted average, and proving the existence of fixed points reduces to showing that the weights concentrate on the nearest pattern. In contrast, the BW-DAM update $\Phi(\xi) = \left( \sum_{i=1}^{N} w_i(\xi) T_i \right)_{\#} \xi$ is a weighted average of optimal transport maps, each of which depends on both the query and the target distribution. Proving the existence of fixed points of $\Phi$ requires bounding $W_2(\Phi(\xi), \Phi(\eta))$, which involves separately controlling how the means and covariances of $\Phi(\xi)$ and $\Phi(\eta)$ differ as the query measures varies. This analysis is carried out in Lemmas 3 and 4 in the Appendix. Moreover, unlike the Euclidean case where exponential storage capacity was proved only for a fixed $\beta$ ($\beta = 1$), Theorem 1 establishes exponential capacity for any $\beta \geq \mathsf{max}\{1, \beta_0\}$.*

**Remark 2.** *To prove Theorem 1, we set the basin radius in Definition 1 to $r = \sqrt{\lambda_{\mathsf{min}}}$, a choice that yields basins whose size is independent of the number of stored Gaussian measures $N$. This contrasts sharply with the Euclidean setting of (Ramsauer et al., 2020), where the basin radius scales as $1/\sqrt{\beta N}$ and thus shrinks as more Gaussians are stored. In particular, when $N$ is exponential in $d$, the Euclidean basins become exponentially (in dimension $d$) small, whereas our basins remain of constant size. This represents an improvement in the capacity-robustness tradeoff:*

*BW-DAM achieves exponential storage capacity while maintaining retrieval robustness that does not degrade with the number of stored Gaussian measures.*

**Remark 3.** *We analyze the per-iteration cost of Algorithm 1 in terms of dimension $d$ and the number of stored Gaussian measures $N$. The square roots $\Sigma_i^{1/2}$ of the stored covariances are fixed across iterations and are computed once, at a one-time cost of $O(Nd^3)$. Within each iteration, Steps 1 and 3 require, for every stored Gaussian measure $X_i$, the matrix $\Sigma_i^{1/2} \Omega \Sigma_i^{1/2}$, formed by two $d \times d$ matrix multiplications costing $O(d^3)$ each under standard matrix multiplication. A single symmetric eigendecomposition of the $d \times d$ symmetric positive-definite matrix $\Sigma_i^{1/2} \Omega \Sigma_i^{1/2}$, computable in $O(d^3)$ via, e.g., the symmetric QR algorithm, simultaneously yields $\left( \Sigma_i^{1/2} \Omega \Sigma_i^{1/2} \right)^{1/2}$ (needed for $D_i = W_2^2(X_i, \xi)$ in Step 1) and $\left( \Sigma_i^{1/2} \Omega \Sigma_i^{1/2} \right)^{-1/2}$ (needed for the transport coefficient $A_i$ in Step 3); The two further multiplications by $\Sigma_i^{1/2}$ that form $A_i$ add $O(d^3)$. Summed over the $N$ stored patterns, Steps 1 and 3 cost $O(Nd^3)$. The remaining operations are cheaper: the softmax weights $\{w_i\}$ cost $O(N)$, the weighted mean $m' = \sum_i w_i \mu_i$ costs $O(Nd)$, the weighted transport coefficient $\tilde{A} = \sum_i w_i A_i$ costs $O(Nd^2)$, and the covariance update $\Omega' = \tilde{A} \Omega \tilde{A}^T$ costs $O(d^3)$, contributing $O(Nd^2 + d^3)$ in total. Adding these contributions, the per-iteration complexity of Algorithm 1 is $O(Nd^3)$.*

## 3.2. Retrieval Guarantees

Having established storage capacity, we turn to retrieval. Algorithm 1 can be used iteratively to retrieve the stored Gaussian distribution given a query Gaussian distribution within a Wasserstein ball around the stored Gaussian distribution.

Algorithm 1 operationalizes the BW-DAM update rule in four steps. First, it computes the pairwise Wasserstein distances between the query measure and each stored Gaussian using the Bures–Wasserstein metric. These distances are then used to construct softmax weights that quantify the influence of each stored pattern. Next, the algorithm determines the optimal transport maps for the covariance matrices, represented by the matrices $A_i$, which specify the linear transformation required to align each stored covariance with the query. Finally, both the mean and covariance of the query measure are updated via a weighted combination of these transport maps, effectively performing a single-step retrieval towards the attractor associated with the most relevant stored pattern.

This procedure can be viewed as a natural generalization of the simple weighted averaging used in the Euclidean dense associative memory case, extended to a transport-based aggregation that respects the geometry of probability

**Algorithm 1** One step of BW-DAM update ($\Phi$ operator)

---

**Require:** Current state $\xi = \mathcal{N}(m, \Omega)$, stored Gaussian measures $\{X_i\}_{i=1}^N$, temperature parameter $\beta$
**Ensure:** Updated state $\xi' = \Phi(\xi) = \mathcal{N}(m', \Omega')$
 1: **Step 1: Compute Wasserstein distances to all stored patterns**
 2: **for** $i = 1$ to $N$ **do**
 3: $\quad D_i \leftarrow \|\mu_i - m\|^2 + \mathsf{tr}(\Sigma_i + \Omega - 2(\Sigma_i^{1/2}\Omega\Sigma_i^{1/2})^{1/2})$
 4: **end for**
 5: **Step 2: Compute softmax weights**
 6: **for** $i = 1$ to $N$ **do**
 7: $\quad w_i \leftarrow \frac{\exp(-\beta D_i)}{\sum_{j=1}^N \exp(-\beta D_j)}$
 8: **end for**
 9: **Step 3: Compute transport map coefficients**
10: **for** $i = 1$ to $N$ **do**
11: $\quad A_i \leftarrow \Sigma_i^{1/2}(\Sigma_i^{1/2}\Omega\Sigma_i^{1/2})^{-1/2}\Sigma_i^{1/2}$
12: **end for**
13: **Step 4: Update means and covariances**
14: $m' \leftarrow \sum_{i=1}^N w_i\mu_i$
15: $\tilde{A} \leftarrow \sum_{i=1}^N w_i A_i$
16: $\Omega' \leftarrow \tilde{A}\Omega\tilde{A}^T$
17: Return the Gaussian measure $\xi' = \mathcal{N}(m', \Omega')$

---

distributions. Concretely, the energy functional $E$ in (1) induces the following *Wasserstein gradient* at a Gaussian measure $\mu \in \mathcal{P}_2(\mathbb{R}^d)$:

$$\nabla_{\mathcal{W}} E(\xi)(x) = 2\sum_{i=1}^N w_i(\xi)\big(T_i(x) - x\big)$$
$$= 2\sum_{i=1}^N w_i(\xi)\Big(\mu_i + A_i(x - m) - x\Big),$$

where $A_i = \Sigma_i^{1/2}(\Sigma_i^{1/2}\Omega\Sigma_i^{1/2})^{-1/2}\Sigma_i^{1/2}$ and the weights $w_i(\xi)$ are as in (2) which has an explicit form due to closed form availability of the Wasserstein-2 metric between Gaussians. Furthermore, as the Bures-Wasserstein gradient is the projection of the Wasserstein gradient to the tangent space at $\mu$, we can simply set the Wasserstein gradient to zero. Rather than explicitly simulating the Wasserstein gradient flow $\frac{d}{dt}\mu_t = -\nabla_{\mathcal{W}}E(\mu_t)$, Algorithm 1 solves the implicit fixed-point equation $\xi^{\text{new}} = \Phi(\xi)$ with $\Phi(\xi) = \Big(\sum_{i=1}^N w_i(\xi)T_i\Big)_\# \xi$. This approach is motivated by efficiency and stability: solving the implicit equation directly moves the query measure closer to a stationary point of $E$ in a single step, effectively "jumping" to the basin of attraction of the most relevant stored pattern. By contrast, an explicit discretization of the Wasserstein gradient flow may require many small steps and careful tuning of step sizes, while still potentially under-shooting or oscillating around the fixed point.

Next, we establish the rate of convergence in 2-Wasserstein distance for retrieving a stored Gaussian measure from a query within its basin of attraction. Toward that end, Theorem 2 guarantees that iterating the update rule in Algorithm 1 from any query $\xi^{(0)}$ within the basin of attraction of a stored Gaussian yields geometric convergence to a unique fixed point in the same basin of attraction. In particular, achieving $\varepsilon$-accuracy in $W_2$ distance requires at most $O(\log(1/\varepsilon))$ iterations, with the constant depending on the contraction constant $L$ and the radius $r = \sqrt{\lambda_{\min}}$ of the basin.

**Theorem 2** (Retrieval Guarantee)**.** *Let* $\{X_i = \mathcal{N}(\mu_i, \Sigma_i)\}_{i=1}^N$ *be Gaussian measures satisfying the separation condition in Assumption 1, with eigenvalues of* $\Sigma_i$ *in* $[\lambda_{\min}, \lambda_{\max}]$*. Suppose* $2\lambda_{\min}d > 1$ *and* $\beta \geq \max\{1, \beta_0\}$*. Define the Wasserstein ball* $\mathcal{B}_i := \{\nu \in \mathcal{P}_2(\mathbb{R}^d) : W_2(X_i, \nu) \leq \sqrt{\lambda_{\min}}\}$*. Then for any query* $\xi^{(0)} \in \mathcal{B}_i$*, the iterates* $\xi^{(k+1)} = \Phi(\xi^{(k)})$ *satisfy:*

1. *$\xi^{(k)} \in \mathcal{B}_i$ for all $k \geq 0$.*

2. *There exists a unique fixed point $X_i^* \in \mathcal{B}_i$ such that*

 $$W_2(\xi^{(k)}, X_i^*) \leq L^k \cdot W_2(\xi^{(0)}, X_i^*) \leq 2\sqrt{\lambda_{\min}} \cdot L^k,$$

 *where $L < 1$ is the contraction constant from Lemma 4.*

3. *Consequently, $\xi^{(k)} \to X_i^*$ as $k \to \infty$, and the number of iterations required to achieve $W_2(\xi^{(k)}, X_i^*) \leq \varepsilon$ is at most $k \geq \frac{\log(2\sqrt{\lambda_{\min}}/\varepsilon)}{\log(1/L)}$.*

Theorem 2 guarantees convergence to a unique fixed point $X_i^*$ within the basin $\mathcal{B}_i$ of the Gaussian measure $X_i$, but does not quantify how close $X_i^*$ is to the actual stored Gaussian measure $X_i$. Indeed $X_i^* = X_i$ only in the limit $\beta \to \infty$; for finite $\beta$, the fixed point generally differs from $X_i$ due to the influence of other stored Gaussian measures. Theorem 3 addresses this gap by showing that after a single application of the map $\Phi$, the retrieval error decays exponentially in dimension $d$.

**Theorem 3** (Retrieval Error)**.** *Let $0 < p < 1$ be a constant, and let $\lambda_{\min}, \lambda_{\max}, \kappa, \alpha, \beta_0$ be as defined in Assumption 1. Consider $N = \lfloor \sqrt{p}e^{\alpha d}\rfloor$ Gaussian measures sampled from a Wasserstein sphere of radius $R = \sqrt{2d\lambda_{\max}(2 + \log\kappa)}$ using Algorithm 2. Assume the separation condition in Assumption 1 holds, $2d\lambda_{\min} > 1$, and $\beta \geq \left\{1, \beta_0, \frac{\alpha}{2\lambda_{\min}}\right\}$. Then for any query $\xi^{(0)} = \mathcal{N}(m, \Omega) \in \mathcal{B}_i$ with eigenvalues of $\Omega$ in $[\lambda_{\min}, \lambda_{\max}]$, the one-step retrieval error decays exponentially in dimension $d$:*

$$W_2(X_i, \Phi(\xi^{(0)})) \leq C\sqrt{d}e^{-\gamma d},$$

*where $C := p^{1/4}\sqrt{\lambda_{\max}[8(2 + \log\kappa) + (3\kappa^2 + 2)]}$ and $\gamma := \beta\lambda_{\min} - \frac{\alpha}{2}$.*

# 4. Numerical Experiments

Having established the theoretical foundations we now turn to empirical validation. We refer to the DAM framework developed in this paper, which operates on the Bures-Wasserstein space using Algorithm 1 as Bures-Wasserstein Dense Associative Memory (BW-DAM). Additional experiments are provided in Section 6.2.

First, we evaluate on real-world image data using CelebA face images. Image pixel values are normalized to the interval $[-1, 1]$ and encoded via a Variational Auto-Encoder (VAE) that outputs a Gaussian $\mathcal{N}(\mu, \Sigma)$ with full covariance in a 32-dimensional latent space. For each trial, we randomly sample 100 CelebA images, encode them as Gaussians, and test retrieval by masking 20% of randomly chosen pixels (set to gray) in the original images. We do not verify whether the encoded Gaussians satisfy the separation condition in Assumption 1, nor whether the masked perturbations lie within the contractive balls of Lemma 4; this experiment probes empirical performance beyond our theoretical assumptions. For BW-DAM, we iterate Algorithm 1 until convergence. For Eu-DAM, we vectorize the Gaussian parameters $(\mu, \Sigma)$ as in the synthetic experiments and iterate until convergence. As an additional baseline, we also consider a pixel-space DAM (Pixel-DAM), which bypasses the Gaussian representation entirely and operates the Euclidean update rule (Ramsauer et al. (2020, Equation 3)) directly on flattened pixel vectors. After convergence, we decode the retrieved Gaussian's mean through the VAE decoder to visualize the result. Results are averaged over five independent trials.

As shown in Figure 3, BW-DAM substantially outperforms both baselines across all values of $\beta$. We attribute this to two factors. First, Eu-DAM treats the Gaussian parameters $(\mu, \Sigma)$ as a flat vector in $\mathbb{R}^{d+d(d+1)/2}$, ignoring the Riemannian structure of the Bures-Wasserstein manifold. Consider two Gaussians $\mathcal{N}(\mu_1, \Sigma_1)$ and $\mathcal{N}(\mu_2, \Sigma_2)$. The squared Euclidean distance between vectorized parameters is $\|\mu_1 - \mu_2\|^2 + \|\Sigma_1 - \Sigma_2\|_F^2$ whereas the squared 2-Wasserstein distance between the two Gaussians is $\|\mu_1 - \mu_2\|^2 + d_B^2(\Sigma_1, \Sigma_2)$, where $d_B$ is the Bures metric. So, while the mean contributions $\|\mu_1 - \mu_2\|^2$ coincide, the Frobenius norm $\|\Sigma_1 - \Sigma_2\|_F$ need not equal the Bures metric $d_B(\Sigma_1, \Sigma_2)$, causing queries to converge to incorrect attractors. Second, Pixel-DAM operates directly on flattened pixel vectors in $\mathbb{R}^d$, where Euclidean distance again does not reflect semantic similarity between images. In contrast, the VAE maps images to a low-dimensional latent space where semantically similar images lie nearby. Consequently, the Gaussian encoding of a masked image remains within the basin of attraction of the original, while the pixel-space encoding need not.

We further evaluate BW-DAM on real-world Gaussian word

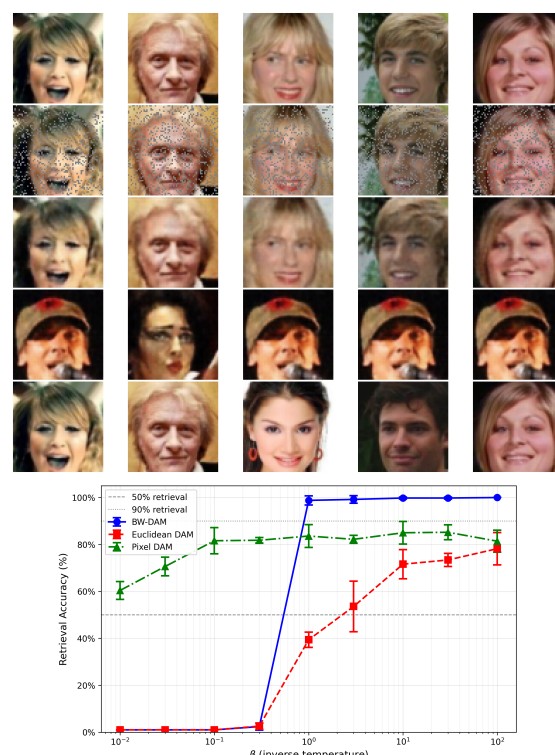

*Figure 3.* Memory retrieval on CelebA images. (Top) Qualitative results for five randomly chosen images with 20% of pixels masked in gray ($\beta = 0.6$). Rows from top to bottom: (1) original images, (2) masked images, (3) BW-DAM retrieval, (4) Eu-DAM retrieval, (5) Pixel-DAM retrieval. (Bottom) Retrieval accuracy as a function of inverse temperature $\beta$ for BW-DAM, Eu-DAM, and Pixel-DAM, evaluated on 100 randomly chosen images. Error bars indicate standard deviation over 5 trials of 100 randomly chosen images each.

embeddings trained using Word2Gauss (Vilnis & McCallum, 2014) on the text8 corpus, which contains approximately 17 million tokens from Wikipedia. After filtering words with fewer than 100 occurrences, we obtain a vocabulary of $N = 11,815$ words. Each word is represented as a 50 dimensional Gaussian distribution with diagonal covariance, trained using the KL-divergence energy function. To test retrieval, we perturb each word's Gaussian embedding by a $W_2$ distance of $\sqrt{\lambda_{\min}}$, where $\lambda_{\min}$ is the minimum eigenvalue across all covariance matrices, yielding a query distribution $\xi_i$. We then apply Algorithm 1 iteratively until convergence, and check whether the nearest (in $W_2$ distance) Gaussian to the final iterate matches the original word. The results in Figure 4b demonstrate a phase transition: for small $\beta$, retrieval accuracy remains low, while for large $\beta$, accuracy approaches 100%, confirming that sufficiently large $\beta$ is necessary to create well-separated energy basins for successful retrieval. We do not verify whether the Gaussian embeddings satisfy the separation condition in Assumption 1; we conjecture that retrieval failures at high

| Original | Iter 0 | Iter 1 | Iter 2 | Iter 3 |
|----------|--------|--------|--------|--------|
| the anarchism abuse | at anarchism abuse | the anarchism abuse | the anarchism abuse | the anarchism abuse |

*(a)* Word evolution at $\beta = 10$

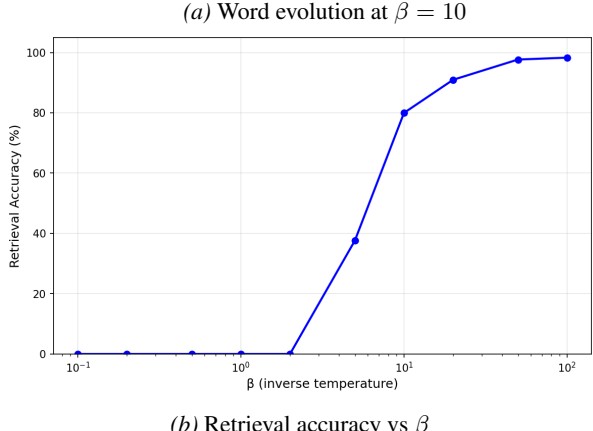

*(b)* Retrieval accuracy vs $\beta$

*Figure 4.* BW-DAM retrieval on Word2Gauss embeddings trained on text8 corpus. Each word's Gaussian embedding is perturbed by $W_2$ distance $\sqrt{\lambda_{\min}}$ and recovered using Algorithm 1. (a) Word evolution showing the nearest word to the current iterate at each step for $\beta = 10$; Iter 0 corresponds to the perturbed query before any updates. (b) Retrieval accuracy as a function of inverse temperature $\beta$.

$\beta$ are due to some Gaussian pairs violating this condition.

## 5. Conclusion

In this work, we extended dense associative memories from the Euclidean space to the Bures–Wasserstein space. We proposed a Wasserstein-energy-based memory, derived explicit retrieval maps, and established theoretical guarantees including high-probability retrieval bounds, and exponential storage capacity. Empirically, our Gaussian DAM achieves robust retrieval under perturbations, demonstrating the utility of transport-based aggregation. Conceptually, this framework enables principled reasoning over probability distributions rather than vectors in the Euclidean space, bridging classical associative memories with modern distributional representations. Future directions include extending to broader probability distribution families (in particular, point-cloud data represented as empirical measures), developing particle-based retrieval algorithms, and exploring applications in generative modeling and probabilistic reasoning.

## Acknowledgments

Krishnakumar Balasubramanian is in part by National Science Foundation (NSF) grant DMS-2413426.

## Impact Statement

This paper presents work whose goal is to advance the field of machine learning. There are many potential societal consequences of our work, none of which we feel must be specifically highlighted here.

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

# 6. Appendix

## 6.1. Related Work

The study of associative memory begins with Hopfield's seminal model (Hopfield, 1982b) which framed memory recall as gradient descent on a quadratic energy landscape. While conceptually foundational, the quadratic Hopfield energy yields only linear storage capacity in the ambient dimension and suffers from spurious attractors at scale. Recent work revived and substantially extended this line of research by introducing highly nonlinear energy functions that dramatically increase capacity. In particular, (Krotov & Hopfield, 2016) proposed LSE style energies and showed that dense associative memories (DAMs) can realize exponentially many stable patterns relative to dimension. Subsequent developments formalized the exponential capacity rigorously and established connections between modern attention/associative recall mechanisms and Hopfield-style energy landscapes (see, for e.g., Demircigil et al. (2017); Ramsauer et al. (2020); Lucibello & Mézard (2024)), showing both practical and theoretical equivalences between attention-like updates and energy-based recall. We also refer the interested reader to recent works (Krotov & Hopfield, 2018; Krotov, 2021; dos Santos et al., 2024; Hoover et al., 2024b; Hu et al., 2023; Hoover et al., 2024a; Wu et al., 2024; Hoover et al., 2025), including the survey by (Krotov et al., 2025) for the state-of-the-art on modern associative memories.

Optimal transport, Wasserstein geometry and barycenters. Optimal transport has emerged as a central tool to compare and interpolate probability measures; the Wasserstein-2 metric, in particular, induces a rich geometric structure that is especially well behaved on Gaussian families (the so-called Bures–Wasserstein geometry). The mathematical theory of Wasserstein barycenters and their computation was significantly advanced by Agueh & Carlier (2011), and efficient numerical algorithms, including entropic regularization approaches, have been developed by Cuturi & Doucet (2014) and others. The computational and theoretical foundations of optimal transport are now well-summarized in recent treatments (Peyré & Cuturi, 2019). Our work leverages these results: stationary points of our distributional LSE energy are self-consistent barycenters in Wasserstein space, and we exploit closed-form formulas and fixed-point iterations available for Gaussian barycenters to derive concrete retrieval dynamics and guarantees.

Very recently, the generative modeling community has increasingly focused on models and architectures that operate over probability distributions. Rectified Point Flow learns continuous velocity fields for point-cloud registration and assembly (Sun et al., 2025), Wasserstein Flow Matching generalizes flow matching to families of distributions via optimal transport geometry (Haviv et al., 2025), and (Bonet et al., 2025) propose flowing measures for distributional generation tasks. These approaches emphasize generative modeling or alignment, whereas our work develops a dense associative memory over probability measures with rigorous capacity and retrieval guarantees, thereby complementing flow-based paradigms. Our approach can be seen as an energy-based generative mechanism in the space of probability measures: fixed points of our Wasserstein LSE energy yield full probability laws that serve as generative attractors. This perspective unifies associative memory and generative modeling, and suggests novel ways to incorporate memory into uncertainty-aware generative pipelines.

## 6.2. Additional Numerical Experiments

### 6.2.1. SYNTHETIC DATA

To further investigate the role of the inverse temperature parameter $\beta$ in high-dimensions, we evaluate the retrieval dynamics of BW-DAM on synthetic Gaussian measures sampled from a Wasserstein sphere of radius $R = \sqrt{2d}$. Specifically, we sample $N = 1000$ Gaussian measures with eigenvalues uniformly distributed in the interval $[0.8, 1.2]$ and mean vectors sampled uniformly on a Euclidean sphere of radius $\sqrt{R^2 - \text{tr}(\Sigma_i)}$, ensuring that each Gaussian measure lies exactly on the Wasserstein sphere of radius $R = \sqrt{2d}$ centered at $\delta_0$. We perturb 750 of the sampled Gaussian measures by a Wasserstein distance of $\sqrt{\lambda_{\min}}$, affecting both the mean and covariance, and run the BW-DAM dynamics to test retrieval. As shown in Figure 5, the results reveal a sharp dependence on $\beta$. For $\beta = 2$, the retrieval is both rapid and accurate: the dynamics converge to the original Gaussian measure in a single iteration across both $d = 10$ and $d = 20$, with negligible variance. In contrast, for $\beta = 0.1$, retrieval fails entirely: the $W_2$ distance increases monotonically, and the dynamics converge to an incorrect Gaussian measure. These findings corroborate our theoretical analysis: sufficiently large $\beta$ ensures that the softmax weights concentrate on the nearest stored pattern, enabling successful retrieval.

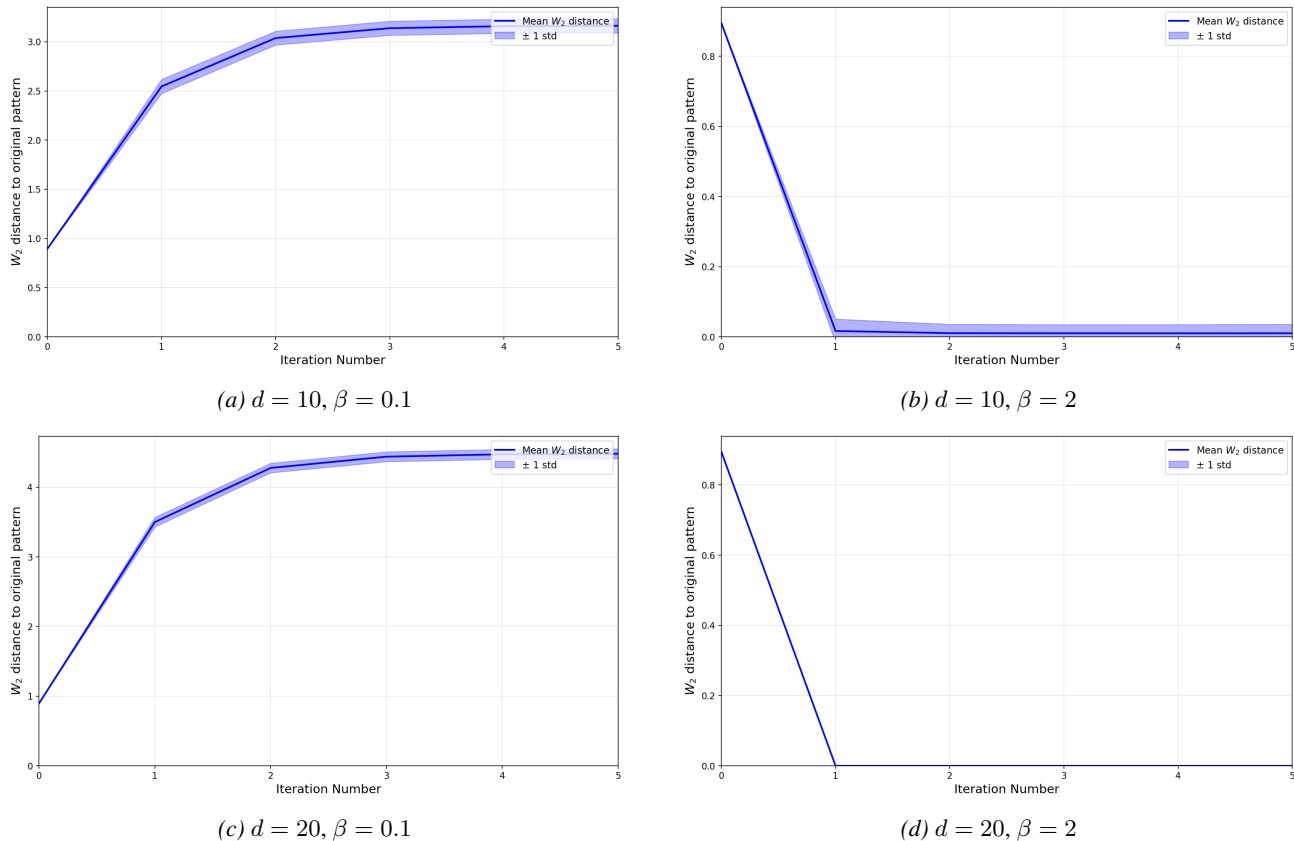

*Figure 5.* Convergence of BW-DAM retrieval dynamics for varying dimension $d$ and temperature $\beta$. We sample $N = 1000$ Gaussians from a Wasserstein sphere of radius $R = \sqrt{2d}$ with eigenvalues in $[0.8, 1.2]$, perturb 75% of the Gaussians to distance $W_2 = \sqrt{\lambda_{\min}}$, and run the dynamics in Algorithm 1. Shaded regions show $\pm 1$ standard deviation.

### 6.2.2. REAL-WORLD DATA

We repeat the experimental setup of Figure 3 on CIFAR-10 images, with results shown in Figure 6. Consistent with the CelebA experiments, BW-DAM substantially outperforms both Eu-DAM and Pixel-DAM across all values of $\beta$. The qualitative results (left panel) demonstrate that BW-DAM successfully retrieves the original images from masked queries, while Eu-DAM baseline converges to incorrect attractors in all the cases. The quantitative results (right panel) show that BW-DAM achieves near-perfect retrieval accuracy for $\beta \geq 1$, whereas Eu-DAM and Pixel-DAM plateau at significantly lower accuracy levels. These findings on a more diverse object recognition dataset further validate that respecting the Bures-Wasserstein geometry of Gaussian embeddings is essential for reliable distributional memory retrieval.

Next, we evaluate BW-DAM on sentence embeddings obtained from GaussCSE (Yoda et al., 2024), a Gaussian embedding model for sentences. We train GaussCSE on 10,000 sentence pairs sampled from the NLI dataset used in SimCSE (Gao et al., 2021), which combines entailment and contradiction pairs from SNLI (Bowman et al., 2015) and MNLI (Williams et al., 2018). The model uses BERT-base-uncased (Devlin et al., 2019) as its backbone and is trained for 3 epochs with a learning rate of $3 \times 10^{-5}$ and batch size of 32. Each sentence is embedded as a Gaussian distribution in $\mathbb{R}^{768}$ with a diagonal covariance matrix. We encode $N = 1000$ unique sentences as stored patterns and compute $\lambda_{\min}$, the smallest eigenvalue across all covariance matrices. For each trial, we sample 100 sentences, perturb their Gaussian embeddings by $W_2$ distance $\sqrt{\lambda_{\min}}$, and run BW-DAM until convergence (i.e., until consecutive iterates differ by less than $\varepsilon = 0.001$ in $W_2$ distance). In Figure 7, we report retrieval accuracy as the fraction of trials where the original sentence is recovered, averaged over 5 independent trials with different random seeds.

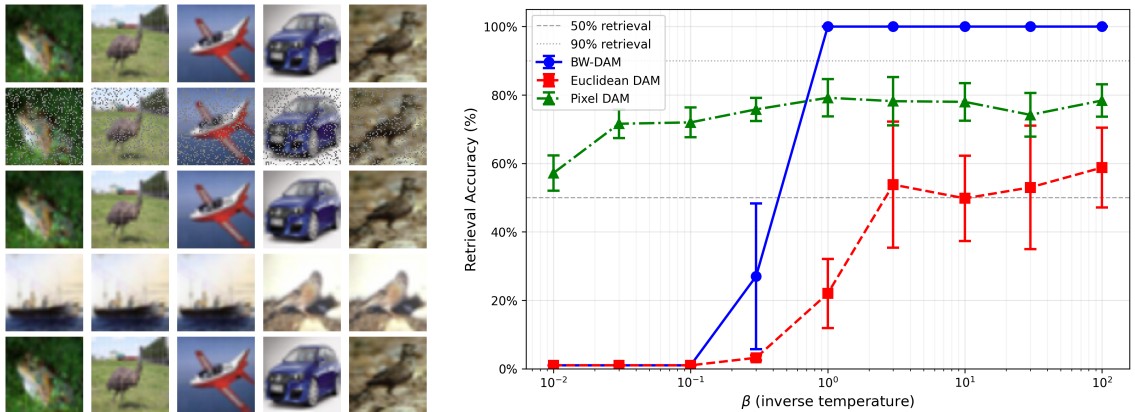

*Figure 6.* Memory retrieval on CIFAR-10 images. (Left) Qualitative results for five randomly chosen images with 20% of pixels masked in gray ($\beta = 0.6$). Rows from top to bottom: (1) original images, (2) masked images, (3) BW-DAM retrieval, (4) Eu-DAM retrieval, (5) Pixel-DAM retrieval. (Right) Retrieval accuracy as a function of inverse temperature $\beta$ for BW-DAM, Eu-DAM, and Pixel-DAM, evaluated on 100 randomly chosen images. Error bars indicate standard deviation over 5 trials of 100 randomly chosen images each.

### 6.3. Preliminary Results

Before presenting the formal results, we provide a visual illustration of the $\Phi$ operator's behavior in Figure 8, which is defined in (4). Panel (a) and (b) shows $W_2(\xi, \Phi(\xi))$ as a heatmap: dark regions near the means of the five Gaussians mark fixed-point neighborhoods, while bright regions indicate strong transformations. Panels (c) and (d) depict the weight functions $w_i(\xi)$, with bright regions where a Gaussian $X_i$ dominates ($w_i(\xi) \approx 1$) and dark regions where other patterns are closer ($w_i(\xi) \approx 0$). Equation 3 has a clear geometric meaning: stationary distributions are precisely those invariant under the weighted barycentric transport field of the stored memories. The fixed point $\xi_*$ is a Wasserstein barycenter with self-consistent weights $w_i(\xi_*)$, ensuring retrieval identifies a distribution that balances the geometric pull of all memories. This implicit barycentric structure directly connects to generative modeling: as in energy-based models, convergence proceeds by descending an energy landscape, but here in Wasserstein space, where attractors are full probability laws satisfying barycentric invariance. Retrieval thus becomes a generative mechanism synthesizing distributions consistent with the stored ensemble, elevating associative memory from pointwise to distributional recall and aligning it with modern generative AI.

In this section, we establish the following three fundamental results which are crucial to prove our results on storage capacity and retrieval rates:

1. The operator $\Phi$ defined by weighted transport maps in (4) preserves Gaussian structure (Lemma 2).
2. For sufficiently separated patterns, $\Phi$ maps Wasserstein balls around patterns to themselves (Lemma 3).
3. Within these balls, the operator $\Phi$ is a contraction, guaranteeing convergence to unique fixed points (Lemma 4).

**Lemma 1.** *Let $X_1 = \mathcal{N}(\mu_1, \Sigma_1)$ and $X_2 = \mathcal{N}(\mu_2, \Sigma_2)$. Then*

$$W_2^2(X_1, X_2) = \|\mu_1 - \mu_2\|^2 + \mathsf{tr}(\Sigma_1 + \Sigma_2 - 2(\Sigma_1^{1/2} \Sigma_2 \Sigma_1^{1/2})^{1/2}).$$

*Proof of Lemma 1.* This is (Lambert et al., 2022)[Equation 5]. $\qquad\square$

**Lemma 2.** *Let $X_i = \mathcal{N}(\mu_i, \Sigma_i)$ for $i = 1, 2, \ldots, N$ be the stored patterns. If $\xi = \mathcal{N}(m, \Sigma_0)$, then*

$$\Phi(\xi) = \mathcal{N}(m', \tilde{A}\Sigma_0\tilde{A}^T),$$

*where $m' = \sum_{i=1}^{N} w_i(\xi)\mu_i$ and $\tilde{A} := \sum_{i=1}^{N} w_i(\xi)A_i$.*

*Proof of Lemma 2.* By Asuka (2011)[Lemma 2.3], the optimal transport map from $\xi = \mathcal{N}(m, \Sigma_0)$ to $X_i = \mathcal{N}(\mu_i, \Sigma_i)$ is given by

$$T_i(x) = \mu_i + A_i(x - m),$$

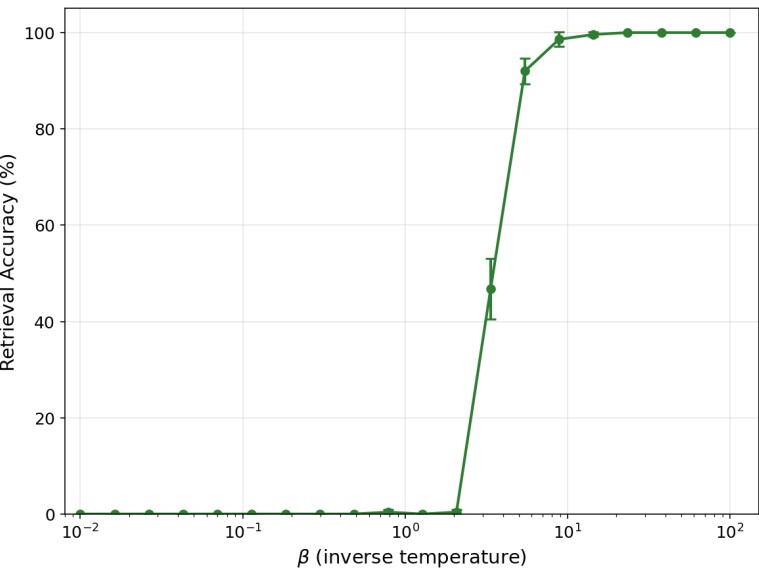

*Figure 7.* Retrieval accuracy of BW-DAM on GaussCSE sentence embeddings as a function of inverse temperature $\beta$. Accuracy is evaluated over 5 trials of 100 randomly chosen sentences each, with perturbations at $W_2$ distance $\sqrt{\lambda_{\min}}$. Error bars indicate standard deviation.

where $A_i = \Sigma_i^{1/2}(\Sigma_i^{1/2}\Sigma_0\Sigma_i^{1/2})^{-1/2}\Sigma_i^{1/2}$. Therefore, the weighted sum of transport maps $T_i$ is:

$$\sum_{i=1}^{N} w_i(\xi)T_i(x) = \sum_{i=1}^{N} w_i(\xi)\left(\mu_i + A_i(x - m)\right)$$
$$= m' + \tilde{A}(x - m),$$

where $\tilde{A} := \sum_{i=1}^{N} w_i(\xi)A_i$ and $m' := \sum_{i=1}^{N} w_i(\xi)\mu_i$. Thus, the weighted sum of transport maps is an affine map. From the definition of operator $\Phi$ in (4), we have

$$\Phi(\xi) = S_{\#}\xi,$$

where $S = \sum_{i=1}^{N} w_i(\xi)T_i$. If $X \sim \xi = \mathcal{N}(m, \Sigma_0)$, then $S(X) = m' + \tilde{A}(X - m)$. Hence, $\mathbb{E}[S(X)] = m'$ and $\mathrm{Var}(S(X)) = \tilde{A}\mathrm{Cov}(X)\tilde{A}^T = \tilde{A}\Sigma_0\tilde{A}^T$ and

$$\Phi(\xi) = \mathcal{N}(m', \tilde{A}\Sigma_0\tilde{A}^T),$$

where $m' = \sum_{i=1}^{N} w_i(\xi)\mu_i$ and $\tilde{A} := \sum_{i=1}^{N} w_i(\xi)A_i$.

$\square$

The idea is to apply Banach's fixed point theorem to prove the existence of a unique fixed point around each pattern. To that end, first we prove that $\Phi$ is a self-map in a neighborhood around each pattern.

**Lemma 3.** *Let $X_1, X_2, \ldots, X_N \in \mathcal{P}_2(\mathbb{R}^d)$ be Gaussian distributions with $X_i = \mathcal{N}(\mu_i, \Sigma_i)$ where the eigenvalues of all $\Sigma_i$ lie in $[\lambda_{\min}, \lambda_{\max}]$. Let $\kappa := \frac{\lambda_{\max}}{\lambda_{\min}}$. Define $M_W := \max_i W_2(X_i, \delta_0)$ where $\delta_0$ is the delta measure at the origin. Assume the separation condition*

$$\Delta_i := \min_{i \neq j}(-\log\langle X_i, X_j\rangle_{L^2}) \geq \frac{d}{2}\log(4\pi\lambda_{\max}) + d.$$

*Assume $d \geq \max\left\{\frac{1}{2\lambda_{\min}}, 4\right\}$, and $\beta \geq \beta_0^{(1)}$, where*

$$\beta_0^{(1)} := \frac{\log N + \log\left(\frac{8M_W^2}{\lambda_{\min}} + 2d\kappa(3\kappa^2 + 2)\right)}{2\lambda_{\min}d}.$$

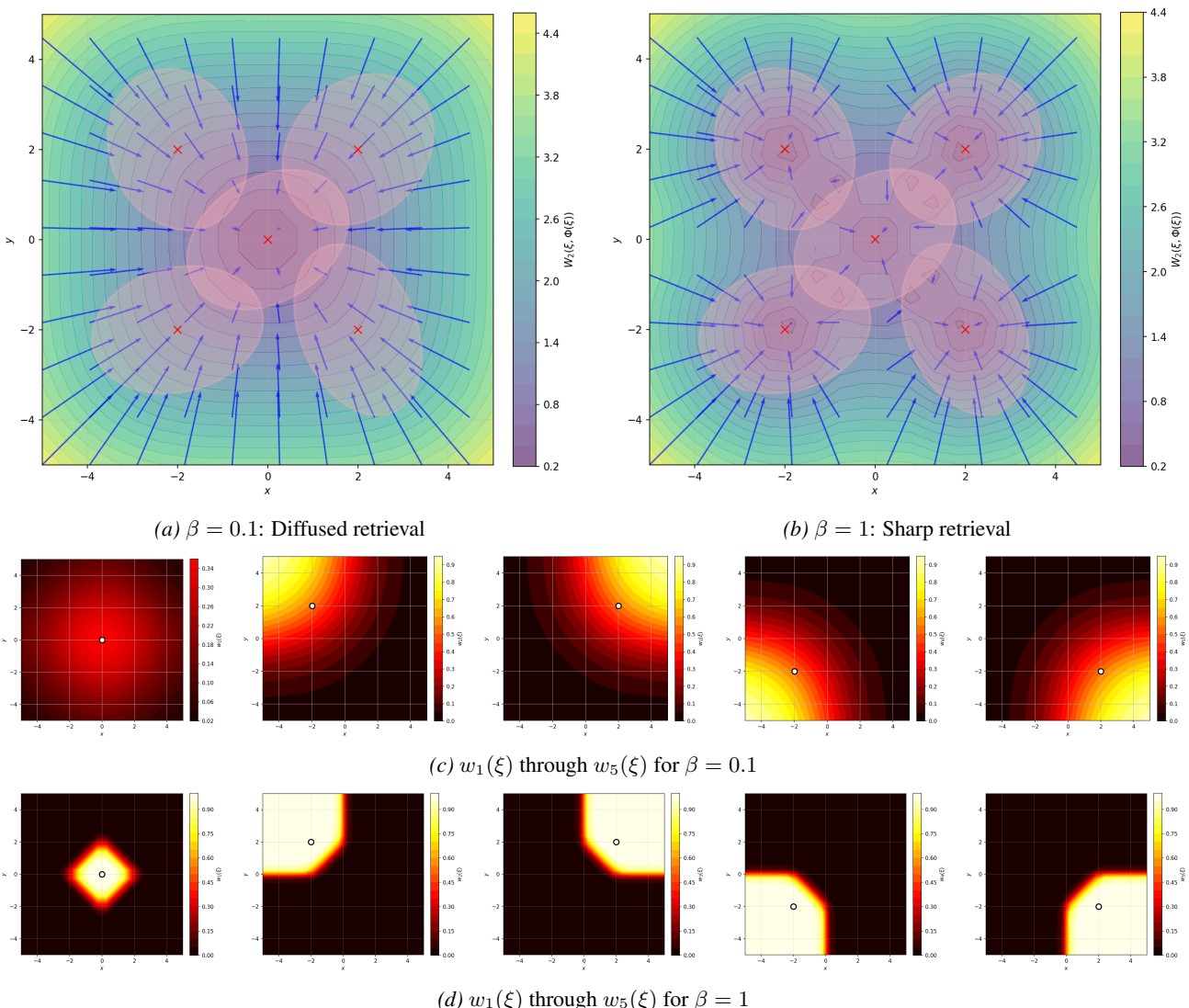

*(a)* $\beta = 0.1$: Diffused retrieval

*(b)* $\beta = 1$: Sharp retrieval

*(c)* $w_1(\xi)$ through $w_5(\xi)$ for $\beta = 0.1$

*(d)* $w_1(\xi)$ through $w_5(\xi)$ for $\beta = 1$

*Figure 8.* Visualization of the $\Phi$ operator and weight functions for two values of the temperature parameter $\beta$. (a,b) Heatmaps show $W_2(\xi, \Phi(\xi))$ computed on a $20 \times 20$ grid and interpolated for smooth visualization. Blue arrows indicate mean displacement vectors from $\xi$ to $\Phi(\xi)$, displayed at every second grid point. Contour lines represent level curves of constant $W_2(\xi, \Phi(\xi))$. Red ellipses show $2\sigma$ contours of stored patterns $X_1, \ldots, X_5$ with means at $(0,0), (\pm 2, \pm 2)$ and anisotropic covariances. (c,d) Weight functions $w_i(\xi)$ showing the influence of each stored pattern across the space, evaluated on the same $20 \times 20$ grid. Query distributions have fixed covariance $0.5I$. Parameters: $N = 5$, with $\beta = 0.1$ (diffused retrieval) and $\beta = 1$ (sharp retrieval).

*Define the radius $r := \sqrt{\lambda_{\min}}$ and the Wasserstein ball $\mathcal{B}_i = \{\nu \in \mathcal{P}_2(\mathbb{R}^d) : W_2(X_i, \nu) \leq r\}$. Let $\xi = \mathcal{N}(m, \Omega) \in \mathcal{B}_i$ with eigenvalues of $\Omega$ in $[\lambda_{\min}, \lambda_{\max}]$. Then*

$$\Phi(\xi) \in \mathcal{B}_i.$$

*Proof of Lemma 3.* By definition of $L^2$ inner product, we have:

$$-\log\langle X_i, X_j\rangle_{L^2} := \frac{d}{2}\log(2\pi) + \frac{1}{2}\log|\Sigma_i + \Sigma_j| + \frac{1}{2}(\mu_i - \mu_j)^T(\Sigma_i + \Sigma_j)^{-1}(\mu_i - \mu_j). \tag{5}$$

Since all eigenvalues of $\Sigma_i, \Sigma_j$ are in $[\lambda_{\min}, \lambda_{\max}]$, the eigenvalues of $\Sigma_i + \Sigma_j$ are in $[2\lambda_{\min}, 2\lambda_{\max}]$, and the eigenvalues of $(\Sigma_i + \Sigma_j)^{-1}$ are in $[1/2\lambda_{\min}, 1/2\lambda_{\max}]$. Thus

$$(2\lambda_{\min})^d \leq |\Sigma_i + \Sigma_j| \leq (2\lambda_{\max})^d, \quad \frac{1}{2\lambda_{\max}}\|\mu_i - \mu_j\|^2 \leq (\mu_i - \mu_j)^T(\Sigma_i + \Sigma_j)^{-1}(\mu_i - \mu_j) \leq \frac{1}{2\lambda_{\min}}\|\mu_i - \mu_j\|^2. \tag{6}$$

By plugging (6) into (5) we get

$$-\log\langle X_i, X_j\rangle_{L^2} \le \frac{d}{2}\log(4\pi\lambda_{\mathsf{max}}) + \frac{1}{4\lambda_{\mathsf{min}}}\|\mu_i - \mu_j\|^2. \tag{7}$$

From the $L^2$ separation condition in the statement of this lemma, we have that for all $i \ne j$,

$$-\log\langle X_i, X_j\rangle_{L^2} \ge \Delta_i \ge \frac{d}{2}\log(4\pi\lambda_{\mathsf{max}}) + d.$$

Therefore, by plugging the bound in (7) into the above inequality, we obtain

$$\|\mu_i - \mu_j\|^2 \ge 4\lambda_{\mathsf{min}}d. \tag{8}$$

In the next steps of the proof, we provide bounds on the weights $w_i(\xi)$, which are used in the operation $\Phi$. From Lemma 1 and the bound in (8), for $i \ne j$:

$$W_2^2(X_i, X_j) = \|\mu_i - \mu_j\|^2 + \mathsf{tr}(\Sigma_i + \Sigma_j - 2(\Sigma_i^{1/2}\Sigma_j\Sigma_i^{1/2})^{1/2}) \ge \|\mu_i - \mu_j\|^2 \ge 4\lambda_{\mathsf{min}}d. \tag{9}$$

For $\xi \in B_i$ and $i \ne j$, by triangle inequality and by using the bound in (9), we obtain:

$$W_2(X_j, \xi) \ge W_2(X_i, X_j) - W_2(X_i, \xi) \ge 2\sqrt{\lambda_{\mathsf{min}}d} - \sqrt{\lambda_{\mathsf{min}}} = \sqrt{\lambda_{\mathsf{min}}}(2\sqrt{d} - 1).$$

By squaring and subtracting $W_2^2(X_i, \xi)$ on both the sides of the above inequality, using $W_2(X_i, \xi) \le r$, and that for $d \ge 4$: $2d - 2\sqrt{d} \ge d$, we get that for $i \ne j$:

$$W_2^2(X_j, \xi) - W_2^2(X_i, \xi) \ge \lambda_{\mathsf{min}}(2\sqrt{d} - 1)^2 - \lambda_{\mathsf{min}} = \lambda_{\mathsf{min}}(4d - 4\sqrt{d}) \ge 2\lambda_{\mathsf{min}}d. \tag{10}$$

Next, by definition of the weight $w_i(\xi)$ and the bound in (10), and by using $\frac{1}{1+x} \ge 1 - x$ for $x > 0$, we obtain

$$\begin{aligned}
w_i(\xi) = \frac{\exp(-\beta W_2^2(X_i, \xi))}{\sum_{k=1}^N \exp(-\beta W_2^2(X_k, \xi))} &= \frac{1}{1 + \sum_{k\ne i}\exp\left(-\beta\left(W_2^2(X_k, \xi) - W_2^2(X_i, \xi)\right)\right)} \\
&\ge \frac{1}{1 + (N-1)e^{-2\beta\lambda_{\mathsf{min}}d}} \\
&\ge 1 - \varepsilon,
\end{aligned} \tag{11}$$

where $\varepsilon := Ne^{-2\beta\lambda_{\mathsf{min}}d}$.

Next, by again using the definition of the weights, the bound in (10), and the definition of $\varepsilon$, we get that for $j \ne i$:

$$\begin{aligned}
w_j(\xi) = \frac{\exp(-\beta W_2^2(X_j, \xi))}{\sum_{k=1}^N \exp(-\beta W_2^2(X_k, \xi))} \le \frac{\exp(-\beta W_2^2(X_j, \xi))}{\exp(-\beta W_2^2(X_i, \xi))} &\le \exp\left(-\beta(W_2^2(X_j, \xi) - W_2^2(X_i, \xi))\right) \\
&\le \exp\left(-2\beta\lambda_{\mathsf{min}}d\right) \\
&= \frac{\varepsilon}{N}.
\end{aligned} \tag{12}$$

The next steps of the proof are toward obtaining an upper bound on $W_2(\Phi(\xi), X_i)$. Using Lemma 1 and Lemma 2, we get

$$W_2^2(\Phi(\xi), X_i) = \|m' - \mu_i\|^2 + \mathsf{tr}(\tilde{A}\Omega\tilde{A}^T + \Sigma_i - 2((\tilde{A}\Omega\tilde{A}^T)^{1/2}\Sigma_i(\tilde{A}\Omega\tilde{A}^T)^{1/2})^{1/2}), \tag{13}$$

where $m' = \sum_{i=1}^N w_i(\xi)\mu_i$ and $\tilde{A} := \sum_{i=1}^N w_i(\xi)A_i$. We refer to the above first term as the *mean error* and the second term as the *covariance error*.

First, we bound the mean error. Using $\|\mu_k\| \le W_2(X_k, \delta_0) \le M_W$ for all $k$, $w_i(\xi) = 1 - \sum_{j\ne i}w_j(\xi)$, Jensen's inequality, and the bound in (11), we get the following bound on the mean error:

$$\|m' - \mu_i\|^2 = \|(w_i(\xi) - 1)\mu_i + \sum_{j\ne i}w_j(\xi)\mu_j\|^2 = \|\sum_{j\ne i}w_j(\xi)(\mu_j - \mu_i)\|^2 \le \sum_{j\ne i}w_j(\xi)\|\mu_j - \mu_i\|^2 \le 4\varepsilon M_W^2. \tag{14}$$

Second, we bound the covariance error. Note that the definition of the transport map $A_j := \Sigma_j^{1/2}(\Sigma_j^{1/2}\Omega\Sigma_j^{1/2})^{-1/2}\Sigma_j^{1/2}$ yields

$$A_j\Omega A_j^T = \Sigma_j^{1/2}(\Sigma_j^{1/2}\Omega\Sigma_j^{1/2})^{-1/2}\Sigma_j^{1/2}\Omega\Sigma_j^{1/2}(\Sigma_j^{1/2}\Omega\Sigma_j^{1/2})^{-1/2}\Sigma_j^{1/2} = \Sigma_j\,, \tag{15}$$

where in the first equality we've used $A_j = A_j^T$. Since $\tilde{A} = \sum_{j=1}^N w_j(\xi)A_j$ and $w_i(\xi) \geq 1-\varepsilon$, we write $\tilde{A} = w_i(\xi)A_i + R$, where $R := \sum_{j\neq i} w_j(\xi)A_j$. Since $A_i, R$ are symmetric, and since $A_i\Omega A_i = \Sigma_i$ from (15), we obtain

$$\begin{aligned}
\tilde{A}\Omega\tilde{A}^T &= (w_i(\xi)A_i + R)\Omega(w_i(\xi)A_i + R)^T \\
&= (w_i(\xi)A_i + R)\Omega(w_i(\xi)A_i + R) \\
&= w_i(\xi)^2 A_i\Omega A_i + w_i(\xi)(A_i\Omega R + R\Omega A_i) + R\Omega R \\
&= w_i(\xi)^2\Sigma_i + w_i(\xi)(A_i\Omega R + R\Omega A_i) + R\Omega R\,.
\end{aligned} \tag{16}$$

Next, we find bounds on operator norms of the matrices in (16). By definition of $A_j$, and since the eigenvalues of $\Sigma_j, \Omega$ lie in the interval $[\lambda_{\mathsf{min}}, \lambda_{\mathsf{max}}]$, we get

$$\begin{aligned}
\|A_j\|_{\mathsf{op}} &= \|\Sigma_j^{1/2}(\Sigma_j^{1/2}\Omega\Sigma_j^{1/2})^{-1/2}\Sigma_j^{1/2}\|_{\mathsf{op}} \\
&\leq \|\Sigma_j^{1/2}\|_{\mathsf{op}}\|(\Sigma_j^{1/2}\Omega\Sigma_j^{1/2})^{-1/2}\|_{\mathsf{op}}\|\Sigma_j^{1/2}\|_{\mathsf{op}} \\
&\leq \sqrt{\lambda_{\mathsf{max}}} \cdot \frac{1}{\lambda_{\mathsf{min}}} \cdot \sqrt{\lambda_{\mathsf{max}}} \\
&=: \kappa\,,
\end{aligned} \tag{17}$$

for all $j \in [N]$, where $\kappa := \frac{\lambda_{\mathsf{max}}}{\lambda_{\mathsf{min}}}$. Using (17), the definition of $R$, and the fact that $w_j(\xi) \leq \frac{\varepsilon}{N-1}$ obtained in (11), we get:

$$\|R\|_{\mathsf{op}} \leq \sum_{j\neq i} w_j(\xi)\|A_j\|_{\mathsf{op}} \leq \varepsilon\kappa\,. \tag{18}$$

Using (18), we get

$$\|R\Omega R\|_{\mathsf{op}} \leq \|R\|_{\mathsf{op}}^2 \cdot \|\Omega\|_{\mathsf{op}} \leq \varepsilon^2\kappa^2\lambda_{\mathsf{max}}\,, \quad \|A_i\Omega R\|_{\mathsf{op}} \leq \|A_i\|_{\mathsf{op}} \cdot \|\Omega\|_{\mathsf{op}} \cdot \|R\|_{\mathsf{op}} \leq \varepsilon\kappa^2\lambda_{\mathsf{max}}\,. \tag{19}$$

By plugging the bounds in (19) into (16), and using $w_i(\xi) \leq 1$, we get

$$\|\tilde{A}\Omega\tilde{A}^T - w_i^2(\xi)\Sigma_i\|_{\mathsf{op}} \leq 2w_i(\xi)\varepsilon\kappa^2\lambda_{\mathsf{max}} + \varepsilon^2\kappa^2\lambda_{\mathsf{max}} \leq \kappa^2\lambda_{\mathsf{max}}(2\varepsilon + \varepsilon^2)\,, \tag{20}$$

and

$$\|\tilde{A}\Omega\tilde{A}^T\|_{\mathsf{op}} \leq \lambda_{\mathsf{max}} + \kappa^2\lambda_{\mathsf{max}}(2\varepsilon + \varepsilon^2)\,. \tag{21}$$

Next, since $w_i(\xi) \geq 1-\varepsilon$:

$$\|w_i^2(\xi)\Sigma_i - \Sigma_i\|_{\mathsf{op}} = |1 - w_i^2(\xi)|\|\Sigma_i\|_{\mathsf{op}} \leq 2\varepsilon\lambda_{\mathsf{max}}\,.$$

Using the above inequality, along with the bound in (20), by applying triangle inequality, using the bound in (21), and using $\varepsilon < 1$, we get

$$\|\tilde{A}\Omega\tilde{A}^T - \Sigma_i\|_{\mathsf{op}} \leq \kappa^2\lambda_{\mathsf{max}}(2\varepsilon + \varepsilon^2) + 2\varepsilon\lambda_{\mathsf{max}} < \lambda_{\mathsf{max}}\varepsilon(3\kappa^2 + 2)\,. \tag{22}$$

Finally, we bound the covariance error. By definition of the Frobenius norm in (Bhatia et al., 2019)[Theorem 1],

$$\left(\mathsf{tr}(\tilde{A}\Omega\tilde{A}^T + \Sigma_i - 2((\tilde{A}\Omega\tilde{A}^T)^{1/2}\Sigma_i(\tilde{A}\Omega\tilde{A}^T)^{1/2})^{1/2})\right)^{1/2} = \min_{U\in\mathcal{U}(d)}\|(\tilde{A}\Omega\tilde{A}^T)^{1/2} - (\Sigma_i)^{1/2}U\|_{\mathsf{F}}\,,$$

where $\mathcal{U}(d)$ is the group of all $d \times d$ unitary matrices. Taking $U = I$ yields

$$\mathsf{tr}(\tilde{A}\Omega\tilde{A}^T + \Sigma_i - 2((\tilde{A}\Omega\tilde{A}^T)^{1/2}\Sigma_i(\tilde{A}\Omega\tilde{A}^T)^{1/2})^{1/2}) \leq \|(\tilde{A}\Omega\tilde{A}^T)^{1/2} - (\Sigma_i)^{1/2}\|_{\mathsf{F}}^2\,. \tag{23}$$

Next, by using the fact that for any two $d$-dimensional positive definite matrices $P, Q$: $\|P - Q\|_{\mathrm{F}} \leq \sqrt{d}\|P - Q\|_{\mathrm{op}}$, using $\|P^{1/2} - Q^{1/2}\|_{\mathrm{op}} \leq \|P - Q\|_{\mathrm{op}}^{1/2}$ for any two positive definite matrices $P, Q$ from (Bhatia, 2013)[Theorem X.1.1], and using the bound in (22), we get

$$
\begin{aligned}
\|(\tilde{A}\Omega\tilde{A}^T)^{1/2} - \Sigma_i^{1/2}\|_{\mathrm{F}}^2 &\leq d\|(\tilde{A}\Omega\tilde{A}^T)^{1/2} - \Sigma_i^{1/2}\|_{\mathrm{op}}^2 \\
&\leq d\|\tilde{A}\Omega\tilde{A}^T - \Sigma_i\|_{\mathrm{op}} \\
&\leq d(3\kappa^2 + 2)\varepsilon\lambda_{\mathsf{max}}.
\end{aligned}
\tag{24}
$$

By plugging the bound in (24) into (23) gives the following bound on the covariance error:

$$
\mathsf{tr}(\tilde{A}\Omega\tilde{A}^T + \Sigma_i - 2((\tilde{A}\Omega\tilde{A}^T)^{1/2}\Sigma_i(\tilde{A}\Omega\tilde{A}^T)^{1/2}) \leq d(3\kappa^2 + 2)\varepsilon\lambda_{\mathsf{max}}.
\tag{25}
$$

Finally, adding the bound on the mean error in (14) and the bound on the covariance error in (25), we obtain the following bound on the 2-Wasserstein distance:

$$
W_2^2(\Phi(\xi), X_i) \leq 4\varepsilon M_W^2 + d(3\kappa^2 + 2)\varepsilon\lambda_{\mathsf{max}}.
\tag{26}
$$

Since $\beta \geq \dfrac{\log N + \log\left(\frac{8M_W^2}{\lambda_{\min}} + \frac{2d\lambda_{\mathsf{max}}(3\kappa^2+2)}{\lambda_{\min}}\right)}{2\lambda_{\min}d}$ from the assumption in this lemma, we have

$$
\varepsilon = Ne^{-2\beta\lambda_{\min}d} \leq \frac{\lambda_{\min}}{8M_W^2}.
\tag{27}
$$

Therefore, from (27), we obtain

$$
4\varepsilon M_W^2 \leq \frac{\lambda_{\min}}{2}.
\tag{28}
$$

Next, note that $\beta \geq \dfrac{\log N + \log\left(\frac{8M_W^2}{\lambda_{\min}} + \frac{2d\lambda_{\mathsf{max}}(3\kappa^2+2)}{\lambda_{\min}}\right)}{2\lambda_{\min}d}$ also implies

$$
\beta \geq \frac{\log N + \log\left(\frac{2d\lambda_{\mathsf{max}}(3\kappa^2+2)}{\lambda_{\min}}\right)}{2\lambda_{\min}d},
$$

and $\varepsilon = Ne^{-2\beta\lambda_{\min}d} \leq \frac{\lambda_{\min}}{2d(3\kappa^2+2)\lambda_{\mathsf{max}}}$. Hence,

$$
d(3\kappa^2 + 2)\varepsilon\lambda_{\mathsf{max}} \leq \frac{\lambda_{\min}}{2}.
\tag{29}
$$

By plugging (28) and (29) into (26), we get

$$
W_2^2(\Phi(\xi), X_i) \leq \lambda_{\min} = r.
$$

This finishes the proof. $\qquad\square$

The following lemma proves a contractive property for the operator $\Phi$.

**Lemma 4.** *Let $X_1, X_2, \ldots, X_N \in \mathcal{P}_2(\mathbb{R}^d)$ be Gaussian distributions with $X_i = \mathcal{N}(\mu_i, \Sigma_i)$ where the eigenvalues of all $\Sigma_i$ lie in $[\lambda_{\min}, \lambda_{\mathsf{max}}]$. Let $\kappa := \frac{\lambda_{\mathsf{max}}}{\lambda_{\min}}$. Define $M_W := \max_i W_2(X_i, \delta_0)$ where $\delta_0$ is the delta measure at the origin. Assume the separation condition*

$$
\Delta_i := \min_{i \neq j}(-\log\langle X_i, X_j\rangle_{L^2}) \geq \frac{d}{2}\log(4\pi\lambda_{\mathsf{max}}) + d.
$$

*Assume $d \geq \max\left\{36, \frac{1}{2\lambda_{\min}}, \frac{(3\kappa^2+2)^2}{288e^4\kappa^{10}}\right\}$ and that $\beta \geq \max\{1, \beta_0^{(2)}\}$, where*

$$
\beta_0^{(2)} = \frac{\log N + \log\left(\left(4M_W + \frac{24\kappa\lambda_{\mathsf{max}}\sqrt{d}}{\sqrt{2}\lambda_{\min}}\right)\left(4M_W + 4\kappa(3+\sqrt{d})\left(3\sqrt{\lambda_{\min}} + 2M_W\right)\right) + \frac{1}{\sqrt{2}\lambda_{\min}}\left(\frac{48\sqrt{d}\kappa\lambda_{\mathsf{max}}^5}{\lambda_{\min}^{9/2}} + \frac{48\kappa^2\lambda_{\mathsf{max}}}{\sqrt{\lambda_{\min}}}\right)\right) + 2}{2\lambda_{\min}d - 1}.
\tag{30}
$$

*Define the radius $r := \sqrt{\lambda_{\min}}$. Then for any $\xi, \eta \in B_r(X_i)$, where $\xi = \mathcal{N}(m_\xi, \Sigma_\xi)$ and $\eta = \mathcal{N}(m_\eta, \Sigma_\eta)$ with eigenvalues in $[\lambda_{\min}, \lambda_{\max}]$, the operator $\Phi$ satisfies*

$$W_2(\Phi(\xi), \Phi(\eta)) \leq L W_2(\xi, \eta),$$

*where $L < 1$.*

*Proof.* By definition of $L^2$ inner product, we have:

$$-\log\langle X_i, X_j\rangle_{L^2} := \frac{d}{2}\log(2\pi) + \frac{1}{2}\log|\Sigma_i + \Sigma_j| + \frac{1}{2}(\mu_i - \mu_j)^T(\Sigma_i + \Sigma_j)^{-1}(\mu_i - \mu_j). \tag{31}$$

Since all eigenvalues of $\Sigma_i, \Sigma_j$ are in $[\lambda_{\min}, \lambda_{\max}]$, the eigenvalues of $\Sigma_i + \Sigma_j$ are in $[2\lambda_{\min}, 2\lambda_{\max}]$, and the eigenvalues of $(\Sigma_i + \Sigma_j)^{-1}$ are in $[1/2\lambda_{\min}, 1/2\lambda_{\max}]$. Thus

$$(2\lambda_{\min})^d \leq |\Sigma_i + \Sigma_j| \leq (2\lambda_{\max})^d, \ \frac{1}{2\lambda_{\max}}\|\mu_i - \mu_j\|^2 \leq (\mu_i - \mu_j)^T(\Sigma_i + \Sigma_j)^{-1}(\mu_i - \mu_j) \leq \frac{1}{2\lambda_{\min}}\|\mu_i - \mu_j\|^2. \tag{32}$$

By plugging (32) into (31) we get

$$-\log\langle X_i, X_j\rangle_{L^2} \leq \frac{d}{2}\log(4\pi\lambda_{\max}) + \frac{1}{4\lambda_{\min}}\|\mu_i - \mu_j\|^2. \tag{33}$$

From the separation condition, we have that for all $i, j \in [N]$,

$$-\log\langle X_i, X_j\rangle_{L^2} \geq \Delta_i \geq \frac{d}{2}\log(4\pi\lambda_{\max}) + d.$$

Therefore, by plugging the bound in (33) into the above inequality, we obtain

$$\|\mu_i - \mu_j\|^2 \geq 4\lambda_{\min}d. \tag{34}$$

Next, for $j \neq i$, $W_2(X_i, X_j) \geq \|\mu_i - \mu_j\| \geq 2\sqrt{\lambda_{\min}d}$. So, for $j \neq i$, by triangle inequality:

$$W_2(X_j, \xi) \geq W_2(X_i, X_j) - W_2(X_i, \xi) \geq 2\sqrt{\lambda_{\min}d} - r = \sqrt{\lambda_{\min}}(2\sqrt{d} - 1),$$

and for $d \geq 4$:

$$W_2^2(X_j, \xi) - W_2^2(X_i, \xi) \geq \lambda_{\min}\left[(2\sqrt{d} - 1)^2 - 1\right] = \lambda_{\min}(4d - 4\sqrt{d}) \geq 2\lambda_{\min}d. \tag{35}$$

Next, we obtain bounds on the weights $w_k(\xi)$ for all $k \in [N]$. By definition of the weight $w_i(\xi)$, by the bound in (35), and by using $\frac{1}{1+x} > 1 - x$ for all $x > 0$, we obtain

$$
\begin{aligned}
w_i(\xi) = \frac{\exp(-\beta W_2^2(X_i, \xi))}{\sum_{k=1}^N \exp(-\beta W_2^2(X_k, \xi))} &= \frac{1}{1 + \sum_{k \neq i}\exp\left(-\beta\left(W_2^2(X_k, \xi) - W_2^2(X_i, \xi)\right)\right)} \\
&\geq \frac{1}{1 + (N-1)\exp(-2\beta\lambda_{\min}d)} \\
&\geq 1 - Ne^{-2\beta\lambda_{\min}d} \\
&= 1 - \varepsilon,
\end{aligned}
\tag{36}
$$

where $\varepsilon := Ne^{-2\beta\lambda_{\min}d}$. Note that by the constraint on $\beta$ in the statement of this lemma, we get

$$\beta \geq \frac{\log(N)}{2\lambda_{\min}d - 1} \geq \frac{\log(N)}{2\lambda_{\min}d}.$$

This yields $N < e^{2\beta\lambda_{\min}d}$ and hence $\varepsilon = Ne^{-2\beta\lambda_{\min}d} < 1$.

Similarly, by using the definition of the weight $w_j(\xi)$, for $j \neq i$, and the bound in (35), we get

$$w_j(\xi) = \frac{\exp(-\beta W_2^2(X_j, \xi))}{\sum_{k=1}^N \exp(-\beta W_2^2(X_k, \xi))} \leq \frac{\exp(-\beta W_2^2(X_j, \xi))}{\exp(-\beta W_2^2(X_i, \xi))} \leq \exp\left(-\beta(W_2^2(X_j, \xi) - W_2^2(X_i, \xi))\right)$$

$$\leq e^{-2\beta\lambda_{\min}d}$$

$$= \frac{\varepsilon}{N}. \tag{37}$$

Now, since $w_j(\xi) = \frac{\exp(-\beta W_2^2(X_j, \xi))}{\sum_{k=1}^N \exp(-\beta W_2^2(X_k, \xi))}$ and $W_2^2(X_j, \xi) = \|\mu_j - m_\xi\|^2 + \text{tr}(\Sigma_j + \Sigma_\xi - 2(\Sigma_\xi^{1/2}\Sigma_j\Sigma_\xi^{1/2})^{1/2})$, we get that the gradient of $w_j(\xi)$ with respect to $m_\xi$ is

$$\nabla_{m_\xi} w_j(\xi) = w_j(\xi)\nabla_{m_\xi} \log w_j(\xi)$$

$$= 2\beta w_j(\xi)\left[\sum_k w_k(\xi)(m_\xi - \mu_k) - (m_\xi - \mu_j)\right]. \tag{38}$$

By triangle inequality: $\|m_\xi - \mu_j\| \leq \|m_\xi - \mu_i\| + \|\mu_i - \mu_j\| \leq 3\sqrt{\lambda_{\min}d}$. This means $\sum_k w_k(\xi)(m_\xi - \mu_k) \leq \max_k \|m_\xi - \mu_k\| \leq 3\sqrt{\lambda_{\min}d}$. By plugging this bound along with the upper bound in (37) into (38), we obtain that for $j \neq i$:

$$\|\nabla_{m_\xi} w_j(\xi)\| \leq 12\beta w_j(\xi)\sqrt{\lambda_{\min}d} \leq 12\beta e^{-2\beta\lambda_{\min}d}\sqrt{\lambda_{\min}d}. \tag{39}$$

From Lemma 2 for $\Phi(\xi) = \mathcal{N}(m'_\xi, \Sigma'_\xi)$ where $m'_\xi = \sum_{i=1}^N w_i(\xi)\mu_i$ and $\Sigma'_\xi = (\sum_i w_i(\xi)A_i)\Sigma_\xi(\sum_i w_i(\xi)A_i)^T$. Also, from Lemma 1 for Gaussian measures $\Phi(\xi), \Phi(\eta)$, we get

$$W_2^2(\Phi(\xi), \Phi(\eta)) = \|m'_\xi - m'_\eta\|^2 + \text{tr}\left(\Sigma'_\xi + \Sigma'_\eta - 2(\Sigma'^{1/2}_\xi\Sigma'_\eta\Sigma'^{1/2}_\xi)^{1/2}\right). \tag{40}$$

Next, we bound the two terms in (40). For the first term, by definition of $m'_\xi, m'_\eta$, and the fact that $\sum_{k=1}^N w_k(\xi) = 1$, we get

$$m'_\xi - m'_\eta = \sum_{j=1}^N [w_j(\xi) - w_j(\eta)]\mu_j = \sum_{j \neq i} [w_j(\xi) - w_j(\eta)](\mu_j - \mu_i). \tag{41}$$

By (Asuka, 2011)[Lemma 2.3], the optimal transport map from the Gaussian measure $\xi = \mathcal{N}(m_\xi, \Sigma_\xi)$ to the Gaussian measure $\eta = \mathcal{N}(m_\eta, \Sigma_\eta)$ is given by $T(x) = m_\eta + A(x - m_\xi)$, where $A = \Sigma_\eta^{1/2}(\Sigma_\eta^{1/2}\Sigma_\xi\Sigma_\eta^{1/2})^{-1/2}\Sigma_\eta^{1/2}$. Next, the geodesic interpolation (in the Wasserstein space) $\xi_t = ((1-t)I + tT)_{\#}\xi$ is the pushforward of $\xi$ via the map $S_t(x) = ((1-t)I + tA)x + (tm_\eta - tAm_\xi)$. Since the pushforward of a Gaussian $\mathcal{N}(m, \Sigma)$ through an affine map $x \to Cx + d$ is $\mathcal{N}(Cm + d, C\Sigma C^T)$, we get that for $0 \leq t \leq 1$: the geodesic interpolation $\xi_t$ between Gaussian measures $\xi$ and $\eta$ is given by $\xi_t = \mathcal{N}(m_{\xi_t}, \Sigma_{\xi_t})$, where $m_{\xi_t} = (1-t)m_\xi + tm_\eta$ and

$$\Sigma_{\xi_t} = ((1-t)I + tA)\Sigma_\xi((1-t)I + tA)^T$$

$$= (1-t)^2\Sigma_\xi + t(1-t)A\Sigma_\xi + t(1-t)\Sigma_\xi A^T + t^2 A\Sigma_\xi A^T$$

$$= (1-t)^2\Sigma_\xi + 2t(1-t)\text{Sym}(A\Sigma_\xi) + t^2\Sigma_\eta. \tag{42}$$

The next steps of the proof are directed toward obtaining a bound on $|w_j(\xi) - w_j(\eta)|$ using the mean-value theorem so that $m'_\xi - m'_\eta$ can be bounded in (41).

Define $g(t) := \frac{e^{-\beta a_j(t)}}{\sum_{k=1}^N e^{-\beta a_k(t)}}$, where $a_j(t) := W_2^2(X_j, \xi_t)$ and $\xi_t = ((1-t)I + tT)_{\#}\xi$ again be the geodesic interpolation of $\xi, \eta$ in the Wasserstein space, where $T$ is the optimal transport map from $\xi$ to $\eta$. By the mean-value theorem, there exists a $t^* \in [0, 1]$ such that:

$$w_j(\eta) - w_j(\xi) = g(1) - g(0) = g'(t^*). \tag{43}$$

Note that for any $t \in [0, 1]$, using the definition of $g(t)$, and differentiating $g(t)$ with respect to $t$, we get:

$$g'(t) = w_j(\xi_t)\beta \left[ \sum_{k=1}^{N} w_k(\xi_t)a_k'(t) - a_j'(t) \right]$$

$$= w_j(\xi_t)\beta \left[ \sum_{k=1}^{N} w_k(\xi_t)(a_k'(t) - a_j'(t)) \right] . \tag{44}$$

Next, for Gaussian measures $X_k = \mathcal{N}(\mu_k, \Sigma_k)$ and $\xi_t = \mathcal{N}(m_{\xi_t}, \Sigma_{\xi_t})$, by Lemma 1, we have:

$$a_k(t) = W_2^2(X_k, \xi_t) = \|\mu_k - m_{\xi_t}\|^2 + d_B^2(\Sigma_k, \Sigma_{\xi_t}) ,$$

where $d_B^2(\Sigma, \Omega) := \operatorname{tr}(\Sigma + \Omega - 2(\Sigma^{1/2}\Omega\Sigma^{1/2})^{1/2})$ is the squared Bures metric. By differentiating $a_k(t)$, we get

$$a_k'(t) = 2\langle \frac{d}{dt}m_{\xi_t} , m_{\xi_t} - \mu_k \rangle + \frac{d}{dt}d_B^2(\Sigma_k, \Sigma_{\xi_t}) .$$

Since $m_{\xi_t} = (1 - t)m_\xi + tm_\eta$, we have $\frac{d}{dt}m_{\xi_t} = m_\eta - m_\xi$. Plugging this into the above equation yields

$$a_k'(t) = 2\langle m_\eta - m_\xi , m_{\xi_t} - \mu_k \rangle + \frac{d}{dt}d_B^2(\Sigma_k, \Sigma_{\xi_t}) ,$$

and therefore

$$a_k'(t) - a_j'(t) = 2\langle m_\eta - m_\xi , \mu_j - \mu_k \rangle + \frac{d}{dt}[d_B^2(\Sigma_k, \Sigma_{\xi_t}) - d_B^2(\Sigma_j, \Sigma_{\xi_t})] . \tag{45}$$

By plugging (45) into the expression for $g'(t)$ in (44), we obtain:

$$g'(t) = w_j(\xi_t)\beta \sum_{k=1}^{N} w_k(\xi_t) \left[ 2\langle m_\eta - m_\xi , \mu_j - \mu_k \rangle + \frac{d}{dt}[d_B^2(\Sigma_k, \Sigma_{\xi_t}) - d_B^2(\Sigma_j, \Sigma_{\xi_t})] \right]$$

$$= 2w_j(\xi_t)\beta\langle m_\eta - m_\xi , \mu_j - \sum_{k=1}^{N} w_k(\xi_t)\mu_k \rangle + w_j(\xi_t)\beta \sum_{k=1}^{N} w_k(\xi_t)\frac{d}{dt}[d_B^2(\Sigma_k, \Sigma_{\xi_t}) - d_B^2(\Sigma_j, \Sigma_{\xi_t})] . \tag{46}$$

By the triangle inequality, for any $t \in [0, 1]$, using $W_2(\xi_t, \xi) = tW_2(\eta, \xi)$, we get:

$$W_2(\xi_t, X_i) \leq W_2(\xi_t, \xi) + W_2(\xi, X_i) = tW_2(\xi, \eta) + W_2(\xi, X_i) \leq 2tr + r \leq 3r . \tag{47}$$

Next, by triangle inequality and the above bound, we obtain:

$$W_2(X_j, \xi_{t^*}) \geq W_2(X_i, X_j) - W_2(X_i, \xi_{t^*}) \geq \sqrt{\lambda_{\min}}(2\sqrt{d} - 3) .$$

By squaring and using (47) again, we get

$$W_2^2(X_j, \xi_{t^*}) - W_2^2(X_i, \xi_{t^*}) \geq \lambda_{\min}(4d - 12\sqrt{d}) .$$

For $d \geq 36$, we have $\lambda_{\min}(4d - 12\sqrt{d}) \geq 2\lambda_{\min}d$. Since (36), (37) provide bounds on the weights for any $\xi \in B_i$ such that (35) holds, we get that for $j \neq i$:

$$w_i(\xi_{t^*}) \geq 1 - \varepsilon , \; w_j(\xi_{t^*}) \leq \frac{\varepsilon}{N} . \tag{48}$$

By plugging (49) into (46) with $t = t^*$ and for a fixed $j \neq i$ yields

$$g'(t^*) = 2w_j(\xi_{t^*})\beta\langle m_\eta - m_\xi , \mu_j - \sum_{k=1}^{N} w_k(\xi_{t^*})\mu_k \rangle + w_j(\xi_{t^*})\beta \sum_{k=1}^{N} w_k(\xi_{t^*})\frac{d}{dt}[d_B^2(\Sigma_k, \Sigma_{\xi_t}) - d_B^2(\Sigma_j, \Sigma_{\xi_t})]\Big|_{t=t^*}$$

$$\leq \frac{2\varepsilon\beta}{N}\langle m_\eta - m_\xi , \mu_j - \sum_{k=1}^{N} w_k(\xi_{t^*})\mu_k \rangle + \frac{\varepsilon^2\beta}{N^2} \sum_{k \neq i} \frac{d}{dt}[d_B^2(\Sigma_k, \Sigma_{\xi_t}) - d_B^2(\Sigma_j, \Sigma_{\xi_t})]\Big|_{t=t^*}$$

$$+ \frac{\varepsilon\beta}{N}w_i(\xi_{t^*})\frac{d}{dt}[d_B^2(\Sigma_i, \Sigma_{\xi_t}) - d_B^2(\Sigma_j, \Sigma_{\xi_t})]\Big|_{t=t^*} . \tag{49}$$

Now we obtain bounds on the different terms in (49). For the first term, by the Cauchy-Schwarz inequality, using $\|m_\eta - m_\xi\| \leq W_2(\xi, \eta)$, triangle inequality, the definition $M_W = \max_i \|\mu_i\|$, and the bounds on weights in (36), (37), we obtain

$$
\begin{aligned}
\frac{2\varepsilon\beta}{N}\langle m_\eta - m_\xi, \, \mu_j - \sum_{k=1}^N w_k(\xi_{t^*})\mu_k\rangle &\leq \frac{2\varepsilon\beta}{N}\|m_\eta - m_\xi\| \cdot \|\mu_j - \sum_{k=1}^N w_k(\xi_{t^*})\mu_k\| \\
&\leq \frac{2\varepsilon\beta}{N}W_2(\xi, \eta)\|\mu_j - w_i(\xi_{t^*})\mu_i - \sum_{k\neq i} w_k(\xi_{t^*})\mu_k\| \\
&\leq \frac{2\varepsilon\beta}{N}W_2(\xi, \eta)\left[\|w_i(\xi_{t^*})(\mu_j - \mu_i)\| + \|(1 - w_i(\xi_{t^*}))\mu_j - \sum_{k\neq i} w_k(\xi_{t^*})\mu_k\|\right] \\
&\leq \frac{2\varepsilon\beta}{N}W_2(\xi, \eta)\left[2M_W + \varepsilon M_W + (N-1)\cdot\frac{\varepsilon}{N}M_W\right] \\
&\leq \frac{4\varepsilon(1+\varepsilon)\beta M_W}{N}W_2(\xi, \eta).
\end{aligned} \tag{50}
$$

Next, we bound the second and third terms in (49). First, consider $\frac{d}{dt}d_B^2(\Sigma_k, \Sigma_{\xi_t})$. Denote $\frac{d}{dt}\Sigma_{\xi_t}$ for element wise derivative of $\Sigma_{\xi_t}$. By chain rule, cyclic property of trace, and the definition of the optimal transport map $A_k(t)$, from $\xi_t$ to $X_k$, in Algorithm 1, we have

$$
\begin{aligned}
\frac{d}{dt}d_B^2(\Sigma_k, \Sigma_{\xi_t}) &= \frac{d}{dt}\text{tr}\left(\Sigma_k + \Sigma_{\xi_t} - 2(\Sigma_k^{1/2}\Sigma_{\xi_t}\Sigma_k^{1/2})^{1/2}\right) \\
&= \text{tr}\left(\frac{d}{dt}\Sigma_{\xi_t} - (\Sigma_k^{1/2}\Sigma_{\xi_t}\Sigma_k^{1/2})^{-1/2}\Sigma_k^{1/2}\frac{d}{dt}\Sigma_{\xi_t}\Sigma_k^{1/2}\right) \\
&= \text{tr}\left(\frac{d}{dt}\Sigma_{\xi_t} - \Sigma_k^{1/2}(\Sigma_k^{1/2}\Sigma_{\xi_t}\Sigma_k^{1/2})^{-1/2}\Sigma_k^{1/2}\frac{d}{dt}\Sigma_{\xi_t}\right) \\
&= \text{tr}\left(\frac{d}{dt}\Sigma_{\xi_t} - A_k(t)\frac{d}{dt}\Sigma_{\xi_t}\right) \\
&= \text{tr}\left((I - A_k(t))\frac{d}{dt}\Sigma_{\xi_t}\right).
\end{aligned} \tag{51}
$$

Therefore, from (51), we get:

$$
\frac{d}{dt}[d_B^2(\Sigma_k, \Sigma_{\xi_t}) - d_B^2(\Sigma_j, \Sigma_{\xi_t})] = \text{tr}\left((A_j(t) - A_k(t))\frac{d}{dt}\Sigma_{\xi_t}\right). \tag{52}
$$

Differentiating the formula for the covariance matrix of the geodesic interpolation between $\xi$ and $\eta$ in (42), we get

$$
\begin{aligned}
\frac{d}{dt}\Sigma_{\xi_t} &= -2(1-t)\Sigma_\xi + 2(1-2t)\text{Sym}(A\Sigma_\xi) + 2t\Sigma_\eta \\
&= 2t(\Sigma_\eta - \Sigma_\xi) - 2(1-2t)\Sigma_\xi + 2(1-2t)\text{Sym}(A\Sigma_\xi) \\
&= 2t(\Sigma_\eta - \Sigma_\xi) - 2(1-2t)\text{Sym}(\Sigma_\xi) + 2(1-2t)\text{Sym}(A\Sigma_\xi) \\
&= 2t(\Sigma_\eta - \Sigma_\xi) + 2(1-2t)\text{Sym}((A-I)\Sigma_\xi).
\end{aligned} \tag{53}
$$

From (Asuka, 2011)[Lemma 2.3] the optimal transport map from $\xi = \mathcal{N}(m_\xi, \Sigma_\xi)$ to $\eta = \mathcal{N}(m_\eta, \Sigma_\eta)$ is given by $T(x) = m_\eta + A(x - m_\xi)$ where $A = \Sigma_\eta^{1/2}(\Sigma_\eta^{1/2}\Sigma_\xi\Sigma_\eta^{1/2})^{-1/2}\Sigma_\eta^{1/2}$. Therefore, $\Sigma_\eta = A\Sigma_\xi A^T$. we obtain

$$
\Sigma_\eta - \Sigma_\xi = A\Sigma_\xi A^T - \Sigma_\xi = (A-I)\Sigma_\xi + \Sigma_\xi(A-I)^T + (A-I)\Sigma_\xi(A-I)^T. \tag{54}
$$

Next, using the fact that the matrix $A$ is symmetric and $\Sigma_\eta = A\Sigma_\xi A^T$, we get

$$
(\Sigma_\xi^{1/2}A\Sigma_\xi^{1/2})^2 = \Sigma_\xi^{1/2}A\Sigma_\xi A\Sigma_\xi^{1/2} = \Sigma_\xi^{1/2}\Sigma_\eta\Sigma_\xi^{1/2}.
$$

Since $(\Sigma_\xi^{1/2} A \Sigma_\xi^{1/2})^2$ and $\Sigma_\xi^{1/2} \Sigma_\eta \Sigma_\xi^{1/2}$ are symmetric positive definite, they have unique square roots so

$$(\Sigma_\xi^{1/2} A \Sigma_\xi^{1/2}) = \left(\Sigma_\xi^{1/2} \Sigma_\eta \Sigma_\xi^{1/2}\right)^{1/2}. \tag{55}$$

Now by definition of the Bures distance, using (55), the cyclic property of trace, $\Sigma_\xi \succeq \lambda_{\min} I$, and the fact that $A$ is symmetric, we get

$$
\begin{aligned}
d_B^2(\Sigma_\xi, \Sigma_\eta) &= \mathrm{tr}(\Sigma_\xi) + \mathrm{tr}(\Sigma_\eta) - 2\mathrm{tr}\left((\Sigma_\xi^{1/2} \Sigma_\eta \Sigma_\xi^{1/2})^{1/2}\right) \\
&= \mathrm{tr}(\Sigma_\xi) + \mathrm{tr}(A\Sigma_\xi A^T) - 2\mathrm{tr}\left((\Sigma_\xi^{1/2} A \Sigma_\xi^{1/2})\right) \\
&= \mathrm{tr}(\Sigma_\xi) + \mathrm{tr}(A^2\Sigma_\xi) - 2\mathrm{tr}(A\Sigma_\xi) \\
&= \mathrm{tr}\left((A-I)^2\Sigma_\xi\right) \\
&\geq \lambda_{\min}\mathrm{tr}((A-I)^2) \\
&= \lambda_{\min}\|A-I\|_{\mathrm{F}}^2.
\end{aligned} \tag{56}
$$

From (56), we infer

$$\|A - I\|_{\mathrm{F}} \leq \frac{1}{\sqrt{\lambda_{\min}}} d_B(\Sigma_\xi, \Sigma_\eta). \tag{57}$$

By using $\|\Sigma_\xi\|_{\mathrm{F}} = \|\Sigma_\eta\|_{\mathrm{F}} = \lambda_{\max}$, plugging (57) into (54), and using $d_B(\Sigma_\xi, \Sigma_\eta) \leq W_2(\xi, \eta) \leq 2r$, we get

$$
\begin{aligned}
\|\Sigma_\xi - \Sigma_\eta\|_{\mathrm{F}} &\leq 2\lambda_{\max}\|A-I\|_{\mathrm{F}} + \lambda_{\max}\|A-I\|_{\mathrm{F}}^2 \\
&\leq \frac{2\lambda_{\max}}{\sqrt{\lambda_{\min}}} d_B(\Sigma_\xi, \Sigma_\eta) + \frac{\lambda_{\max}}{\lambda_{\min}} d_B^2(\Sigma_\xi, \Sigma_\eta) \\
&\leq \frac{2\lambda_{\max}}{\sqrt{\lambda_{\min}}}\left(1 + \frac{2r}{\sqrt{\lambda_{\min}}}\right) W_2(\xi, \eta) \\
&= \frac{6\lambda_{\max}}{\sqrt{\lambda_{\min}}} W_2(\xi, \eta).
\end{aligned} \tag{58}
$$

By plugging (58) and (57) into (53), we get

$$
\begin{aligned}
\left\|\frac{d}{dt}\Sigma_{\xi_t}\right\|_{\mathrm{F}} &\leq 2t\|\Sigma_\eta - \Sigma_\xi\|_{\mathrm{F}} + 2|1-2t|\,\|\mathrm{Sym}((A-I)\Sigma_\xi)\|_{\mathrm{F}} \\
&\leq \frac{4\lambda_{\max}}{\sqrt{\lambda_{\min}}}\left(1 + \frac{\sqrt{d}}{2}\right) W_2(\xi, \eta) + 2\,\|(A-I)\Sigma_\xi\|_{\mathrm{F}} \\
&\leq \frac{4\lambda_{\max}}{\sqrt{\lambda_{\min}}}\left(1 + \frac{\sqrt{d}}{2}\right) W_2(\xi, \eta) + 2\,\|A-I\|_{\mathrm{F}}\,\|\Sigma_\xi\|_{\mathrm{op}} \\
&\leq \frac{4\lambda_{\max}}{\sqrt{\lambda_{\min}}}\left(1 + \frac{\sqrt{d}}{2}\right) W_2(\xi, \eta) + 2\frac{\lambda_{\max}}{\sqrt{\lambda_{\min}}} W_2(\xi, \eta) \\
&= \frac{2\lambda_{\max}}{\sqrt{\lambda_{\min}}}(3 + \sqrt{d}) W_2(\xi, \eta).
\end{aligned} \tag{59}
$$

By applying trace inequality and plugging (59) into (52), using triangle inequality, plugging (57) into (52), using the definition $M_W := \max_i W_2(\delta_0, X_i)$, using the fact $W_2(\xi_t, X_j) \leq W_2(\xi_t, X_i) + W_2(X_i, X_j) \leq r + 2M_W$, and using the

definition of the radius of the Wasserstein ball $r = \sqrt{\lambda_{\min}}$, we get

$$
\begin{aligned}
\frac{d}{dt}[d_B^2(\Sigma_k, \Sigma_{\xi_t}) - d_B^2(\Sigma_j, \Sigma_{\xi_t})] &= \text{tr}\left((A_j(t) - A_k(t))\frac{d}{dt}\Sigma_{\xi_t}\right) \\
&\leq \|A_j(t) - A_k(t)\|_\text{F} \left\|\frac{d}{dt}\Sigma_{\xi_t}\right\|_\text{F} \\
&\leq [\|A_j(t) - I\|_\text{F} + \|A_k(t) - I\|_\text{F}] \cdot \frac{2\lambda_{\max}}{\sqrt{\lambda_{\min}}}(3 + \sqrt{d})W_2(\xi, \eta) \\
&\leq \frac{1}{\sqrt{\lambda_{\min}}}(d_B(\Sigma_{\xi_t}, \Sigma_j) + d_B(\Sigma_{\xi_t}, \Sigma_k)) \cdot \frac{2\lambda_{\max}}{\sqrt{\lambda_{\min}}}(3 + \sqrt{d})W_2(\xi, \eta) \\
&\leq \frac{2\lambda_{\max}}{\lambda_{\min}}(3 + \sqrt{d})\left(W_2(\xi_t, X_j) + W_2(\xi_t, X_k)\right)W_2(\xi, \eta) \\
&\leq \frac{4\lambda_{\max}}{\lambda_{\min}}(3 + \sqrt{d})(3r + 2M_W)W_2(\xi, \eta) \\
&= \frac{4\lambda_{\max}}{\lambda_{\min}}(3 + \sqrt{d})\left(3\sqrt{\lambda_{\min}} + 2M_W\right)W_2(\xi, \eta).
\end{aligned}
\tag{60}
$$

Note that the above bound holds uniformly for all $t \in [0, 1]$ (including $t^*$). To simplify notation in subsequent steps define $C_1(d) := \frac{4\lambda_{\max}}{\lambda_{\min}}(3 + \sqrt{d})\left(3\sqrt{\lambda_{\min}} + 2M_W\right)$. By plugging the bound in (50) and (60) into (49), and using $\varepsilon < 1$, we get

$$
\begin{aligned}
|g'(t^*)| &\leq \frac{4\varepsilon(1 + \varepsilon)\beta M_W}{N}W_2(\xi, \eta) + \frac{\varepsilon^2\beta C_1(d)}{N}W_2(\xi, \eta) + \frac{\varepsilon\beta C_1(d)}{N}W_2(\xi, \eta) \\
&\leq \frac{8\varepsilon\beta M_W}{N}W_2(\xi, \eta) + \frac{2\varepsilon\beta C_1(d)}{N}W_2(\xi, \eta) \\
&= \frac{2\varepsilon\beta}{N}(4M_W + C_1(d))W_2(\xi, \eta).
\end{aligned}
\tag{61}
$$

By plugging the bound in (61) into (43), we get that for all $j \neq i$:

$$
|w_j(\xi) - w_j(\eta)| \leq |g'(t^*)| \leq \frac{2\varepsilon\beta}{N}(4M_W + C_1(d))W_2(\xi, \eta).
\tag{62}
$$

Next, by plugging the bound in (62) into (41), and using $\|\mu_k\| \leq M_W$ for all $k$, we get the following bound:

$$
\begin{aligned}
\|m'_\xi - m'_\eta\| &\leq \sum_{j \neq i}|w_j(\xi) - w_j(\eta)| \cdot 2M_W \\
&\leq 4\varepsilon\beta M_W(4M_W + C_1(d))W_2(\xi, \eta).
\end{aligned}
\tag{63}
$$

This yields the bound on the first term in (40). The next steps of the proof provide a bound on the second term in (40).

We recall the transport map coefficient $\tilde{A} = \sum_{j=1}^N w_j(\xi)A_j$ from Lemma 2, where $A_j$ is the transport map coefficient from $\xi$ to $X_j$. Since, we have two Gaussian distributions $\xi, \eta$ in this lemma, in order to distinguish between the transport map coefficients corresponding to $\xi, \eta$, we use the notation $\tilde{A}^\xi := \sum_{j=1}^N w_j(\xi)A_j^\xi$ and $\tilde{A}^\eta := \sum_{j=1}^N w_j(\eta)A_j^\eta$, where $A_j^\xi, A_j^\eta$ are the transport map coefficients from $\xi$ and $\eta$ to $X_j$, respectively. Recall the definition $A_k^\xi := \Sigma_k^{1/2}(\Sigma_k^{1/2}\Sigma_\xi\Sigma_k^{1/2})^{-1/2}\Sigma_k^{1/2}$, $A_k^\eta := \Sigma_k^{1/2}(\Sigma_k^{1/2}\Sigma_\eta\Sigma_k^{1/2})^{-1/2}\Sigma_k^{1/2}$. To simplify notation in the next steps, we define $M_k^\xi := \Sigma_k^{1/2}\Sigma_\xi\Sigma_k^{1/2}$ and $M_k^\eta := \Sigma_k^{1/2}\Sigma_\eta\Sigma_k^{1/2}$. Note that

$$
\begin{aligned}
A_k^\xi - A_k^\eta &= \Sigma_k^{1/2}\left[(M_k^\xi)^{-1/2} - (M_k^\eta)^{-1/2}\right]\Sigma_k^{1/2} \\
&= \Sigma_k^{1/2}\left[((M_k^\xi)^{-1})^{1/2} - ((M_k^\eta)^{-1})^{1/2}\right]\Sigma_k^{1/2}.
\end{aligned}
\tag{64}
$$

By applying (Bhatia, 2013)[Theorem X.3.8] with $f(t) = t^{1/2}$ and using $(M_k^\xi)^{-1} \succeq \frac{1}{\lambda_{\max}^2}I$, $(M_k^\eta)^{-1} \succeq \frac{1}{\lambda_{\max}^2}I$, using

$\|\cdot\|_{\mathrm{op}} \leq \|\cdot\|_{\mathrm{F}}$, using $(M_k^\xi)^{-1} \preceq \frac{1}{\lambda_{\max}^2} I$, $(M_k^\eta)^{-1} \preceq \frac{1}{\lambda_{\max}^2} I$, and using the bound in (58), we get

$$
\begin{aligned}
\|((M_k^\xi)^{-1})^{1/2} - ((M_k^\eta)^{-1})^{1/2}\|_{\mathrm{op}} &\leq \frac{\lambda_{\max}}{2} \|(M_k^\xi)^{-1} - (M_k^\eta)^{-1}\|_{\mathrm{op}} \\
&= \frac{\lambda_{\max}}{2} \|(M_k^\xi)^{-1}(M_k^\eta - M_k^\xi)(M_k^\eta)^{-1}\|_{\mathrm{op}} \\
&\leq \frac{\lambda_{\max}}{2} \|(M_k^\xi)^{-1}\|_{\mathrm{op}} \|M_k^\eta - M_k^\xi\|_{\mathrm{op}} \|(M_k^\eta)^{-1}\|_{\mathrm{op}} \\
&\leq \frac{\lambda_{\max}}{2\lambda_{\min}^4} \|M_k^\eta - M_k^\xi\|_{\mathrm{op}} \\
&\leq \frac{\lambda_{\max}}{2\lambda_{\min}^4} \|\Sigma_k^{1/2}(\Sigma_\eta - \Sigma_\xi)\Sigma_k^{1/2}\|_{\mathrm{op}} \\
&\leq \frac{\lambda_{\max}}{2\lambda_{\min}^4} \cdot \|\Sigma_k^{1/2}\|_{\mathrm{op}}^2 \|\Sigma_\eta - \Sigma_\xi\|_{\mathrm{op}} \\
&\leq \frac{\lambda_{\max}^2}{2\lambda_{\min}^4} \cdot \frac{6\lambda_{\max}}{\sqrt{\lambda_{\min}}} W_2(\xi,\eta) \\
&= \frac{3\lambda_{\max}^3}{\lambda_{\min}^{9/2}} W_2(\xi,\eta) \,.
\end{aligned}
\tag{65}
$$

By plugging the bound in (65) into (64), we get

$$
\begin{aligned}
\|A_k^\xi - A_k^\eta\|_{\mathrm{op}} &\leq \|\Sigma_k^{1/2}\|_{\mathrm{op}}^2 \cdot \|((M_k^\xi)^{-1})^{1/2} - ((M_k^\eta)^{-1})^{1/2}\|_{\mathrm{op}} \\
&\leq \frac{3\lambda_{\max}^4}{\lambda_{\min}^{9/2}} W_2(\xi,\eta) \,.
\end{aligned}
\tag{66}
$$

The final steps of the proof provide an upper bound on the second term in (40). We write $\tilde{A}^\xi = A_i^\xi + R^\xi$, where $R^\xi = \sum_{j\neq i} w_j(\xi)(A_j^\xi - A_i^\xi)$. Since $\Sigma_\xi' = \tilde{A}\Sigma_\xi\tilde{A}^T$ and $\Sigma_\eta' = \tilde{A}\Sigma_\eta\tilde{A}^T$, we get

$$
\begin{aligned}
\Sigma_\xi' &= (A_i^\xi + R^\xi)\Sigma_\xi(A_i^\xi + R^\xi)^T \\
&= A_i^\xi\Sigma_\xi(A_i^\xi)^T + A_i^\xi\Sigma_\xi(R^\xi)^T + R^\xi\Sigma_\xi(A_i^\xi)^T + R^\xi\Sigma_\xi(R^\xi)^T \\
&= \Sigma_i + A_i^\xi\Sigma_\xi(R^\xi)^T + R^\xi\Sigma_\xi(A_i^\xi)^T + R^\xi\Sigma_\xi(R^\xi)^T \,,
\end{aligned}
\tag{67}
$$

and

$$
\Sigma_\xi' - \Sigma_\eta' = \left(A_i^\xi\Sigma_\xi(R^\xi)^T - A_i^\eta\Sigma_\eta(R^\eta)^T\right) + \left(R^\xi\Sigma_\xi(A_i^\xi)^T - R^\eta\Sigma_\eta(A_i^\eta)^T\right) + \left(R^\xi\Sigma_\xi(R^\xi)^T - R^\eta\Sigma_\eta(R^\eta)^T\right) \,.
\tag{68}
$$

Note that by triangle inequality and (17),

$$
\|R^\xi\|_{\mathrm{op}} = \left\|\sum_{j\neq i} w_j(\xi)(A_j^\xi - A_i^\xi)\right\|_{\mathrm{op}} \leq \frac{(N-1)\varepsilon}{N}(\|A_j^\xi\|_{\mathrm{op}} + \|A_i^\xi\|_{\mathrm{op}}) \leq 2\varepsilon\kappa \,.
\tag{69}
$$

Also,

$$
(R^\xi)^T = R^\xi \,,
\tag{70}
$$

because $(A_k^\xi)^T = A_k^\xi$ for all $k$. Additionally,

$$
\begin{aligned}
R^\xi - R^\eta &= \sum_{j\neq i} \left[w_j(\xi)(A_j^\xi - A_i^\xi) - w_j(\eta)(A_j^\eta - A_i^\eta)\right] \\
&= \sum_{j\neq i} \left[(w_j(\xi) - w_j(\eta))(A_j^\xi - A_i^\xi)\right] + \sum_{j\neq i} \left[w_j(\eta)\left((A_j^\xi - A_i^\xi) - (A_j^\eta - A_i^\eta)\right)\right] \,.
\end{aligned}
\tag{71}
$$

For the first term in (71), by using (17) and (62), and using $\|A\|_F \le \sqrt{d}\|A\|_{op}$ for any matrix $A$, we obtain

$$\left\| \sum_{j \ne i} \left[ (w_j(\xi) - w_j(\eta))(A_j^\xi - A_i^\xi) \right] \right\|_F \le (N-1) \cdot \frac{2\varepsilon\beta}{N}(4M_W + C_1(d))W_2(\xi,\eta) \cdot 2\sqrt{d}\kappa$$

$$\le 4\varepsilon\beta\sqrt{d}\kappa(4M_W + C_1(d))W_2(\xi,\eta). \tag{72}$$

For the second term in (71), by using $\sum_{j \ne i} w_j(\eta) \le \varepsilon$, the fact that $\|A\|_F \le \sqrt{d}\|A\|_{op}$ for any matrix $A$, and the bound in (66), we get

$$\left\| \sum_{j \ne i} \left[ w_j(\eta) \left( (A_j^\xi - A_i^\xi) - (A_j^\eta - A_i^\eta) \right) \right] \right\|_F \le \sum_{j \ne i} w_j(\eta) \left[ \|(A_j^\xi - A_j^\eta)\|_F + \|(A_i^\xi - A_i^\eta)\|_F \right]$$

$$\le \frac{6\lambda_{max}^4\sqrt{d}}{\lambda_{min}^{9/2}} W_2(\xi,\eta) \sum_{j \ne i} w_j(\eta)$$

$$\le \frac{6\lambda_{max}^4\sqrt{d}\varepsilon}{\lambda_{min}^{9/2}} W_2(\xi,\eta). \tag{73}$$

By plugging (72) and (73) into (71), we get

$$\|R^\xi - R^\eta\|_F \le \left[ 4\varepsilon\beta\sqrt{d}\kappa(4M_W + C_1(d)) + \frac{6\lambda_{max}^4\sqrt{d}\varepsilon}{\lambda_{min}^{9/2}} \right] W_2(\xi,\eta). \tag{74}$$

Now consider the first term in (68). By using (69), (70), (74) along with submultiplicativity of the Frobenius norm, the bounds in (58) and (66), we obtain

$$\|A_i^\xi \Sigma_\xi (R^\xi)^T - A_i^\eta \Sigma_\eta (R^\eta)^T\|_F = \left\| (A_i^\xi - A_i^\eta)\Sigma_\xi (R^\xi)^T + A_i^\eta(\Sigma_\xi - \Sigma_\eta)(R^\xi)^T + A_i^\eta \Sigma_\eta (R^\xi - R^\eta)^T \right\|_F$$

$$\le \|A_i^\xi - A_i^\eta\|_F \|\Sigma_\xi\|_{op} \|(R^\xi)^T\|_{op} + \|A_i^\eta\|_{op} \|\Sigma_\xi - \Sigma_\eta\|_F \|(R^\xi)^T\|_{op}$$

$$+ \|A_i^\eta\|_{op} \|\Sigma_\eta\|_{op} \|(R^\xi - R^\eta)^T\|_F$$

$$\le \frac{6\sqrt{d}\lambda_{max}^5 \varepsilon\kappa}{\lambda_{min}^{9/2}} W_2(\xi,\eta) + \frac{12\varepsilon\lambda_{max}\kappa^2}{\sqrt{\lambda_{min}}} W_2(\xi,\eta)$$

$$+ \kappa\lambda_{max} \left[ 4\varepsilon\beta\sqrt{d}\kappa(4M_W + C_1(d)) + \frac{6\lambda_{max}^4\sqrt{d}\varepsilon}{\lambda_{min}^{9/2}} \right] W_2(\xi,\eta). \tag{75}$$

Next, consider the second term in (68). Since the transpose of the second term in (68) is equal to the first term in (68), we obtain from (75) the following bound

$$\left\| R^\xi \Sigma_\xi (A_i^\xi)^T - R^\eta \Sigma_\eta (A_i^\eta)^T \right\|_F \le \frac{6\sqrt{d}\lambda_{max}^5 \varepsilon\kappa}{\lambda_{min}^{9/2}} W_2(\xi,\eta) + \frac{12\varepsilon\lambda_{max}\kappa^2}{\sqrt{\lambda_{min}}} W_2(\xi,\eta)$$

$$+ \kappa\lambda_{max} \left[ 4\varepsilon\beta\sqrt{d}\kappa(4M_W + C_1(d)) + \frac{6\lambda_{max}^4\sqrt{d}\varepsilon}{\lambda_{min}^{9/2}} \right] W_2(\xi,\eta). \tag{76}$$

Finally consider the third term in (68). By using the bounds in (74), (17), (18), (58), (69), (70), and the fact that

$\|A\|_{\mathrm{F}} \le \sqrt{d}\|A\|_{\mathrm{op}}$ for any matrix $A$, we get

$$
\begin{aligned}
\|R^\xi \Sigma_\xi (R^\xi)^T - R^\eta \Sigma_\eta (R^\eta)^T\|_{\mathrm{F}} &= \|(R^\xi - R^\eta)\Sigma_\xi (R^\xi)^T + R^\eta(\Sigma_\xi - \Sigma_\eta)(R^\xi)^T + R^\eta \Sigma_\eta (R^\xi - R^\eta)^T\|_{\mathrm{F}} \\
&\le \|(R^\xi - R^\eta)\Sigma_\xi (R^\xi)^T\|_{\mathrm{F}} + \|R^\eta(\Sigma_\xi - \Sigma_\eta)(R^\xi)^T\|_{\mathrm{F}} + \|R^\eta \Sigma_\eta (R^\xi - R^\eta)^T\|_{\mathrm{F}} \\
&\le \|R^\xi - R^\eta\|_{\mathrm{F}}\|\Sigma_\xi\|_{\mathrm{op}}\|R^\xi\|_{\mathrm{op}} + \|R^\eta\|_{\mathrm{op}}\|\Sigma_\xi - \Sigma_\eta\|_{\mathrm{F}}\|R^\xi\|_{\mathrm{op}} \\
&\quad + \|R^\eta\|_{\mathrm{op}}\|\Sigma_\eta\|_{\mathrm{op}}\|R^\xi - R^\eta\|_{\mathrm{F}} \\
&\le \left[ 4\varepsilon\beta\sqrt{d}\kappa(4M_W + C_1(d)) + \frac{6\lambda_{\max}^4\sqrt{d}\varepsilon}{\lambda_{\min}^{9/2}} \right] 4\lambda_{\max}\varepsilon\kappa W_2(\xi,\eta) \\
&\quad + \frac{24\varepsilon^2\kappa^2\lambda_{\max}}{\sqrt{\lambda_{\min}}} W_2(\xi,\eta) \,.
\end{aligned}
\tag{77}
$$

By plugging the bounds in (75), (76), (77) into (68), we get

$$
\begin{aligned}
\|\Sigma_\xi' - \Sigma_\eta'\|_{\mathrm{F}} &\le \left[ \frac{12\sqrt{d}\lambda_{\max}^5\varepsilon\kappa}{\lambda_{\min}^{9/2}} + \frac{24\varepsilon\lambda_{\max}\kappa^2}{\sqrt{\lambda_{\min}}} + 2\kappa\lambda_{\max}\left[ 4\varepsilon\beta\sqrt{d}\kappa(4M_W + C_1(d)) + \frac{6\lambda_{\max}^4\sqrt{d}\varepsilon}{\lambda_{\min}^{9/2}} \right] \right] W_2(\xi,\eta) \\
&\quad + \left[ \left[ 4\varepsilon\beta\sqrt{d}\kappa(4M_W + C_1(d)) + \frac{6\lambda_{\max}^4\sqrt{d}\varepsilon}{\lambda_{\min}^{9/2}} \right] 2\lambda_{\max}\varepsilon\kappa + \frac{24\varepsilon^2\kappa^2\lambda_{\max}}{\sqrt{\lambda_{\min}}} \right] W_2(\xi,\eta) \,.
\end{aligned}
\tag{78}
$$

To simplify notation in the next steps, define

$$
C_R(d) := 4\beta\sqrt{d}\kappa(4M_W + C_1(d)) + \frac{6\lambda_{\max}^4\sqrt{d}}{\lambda_{\min}^{9/2}} \,.
$$

$$
C_{\Sigma,1}(d) := \frac{12\sqrt{d}\lambda_{\max}^5\varepsilon\kappa}{\lambda_{\min}^{9/2}} + \frac{24\varepsilon\lambda_{\max}\kappa^2}{\sqrt{\lambda_{\min}}} + 2\kappa\lambda_{\max}C_R(d) \,, \quad C_{\Sigma,2} := 4\kappa\lambda_{\max}C_R(d) + \frac{24\kappa^2\lambda_{\max}}{\sqrt{\lambda_{\min}}} \,.
$$

Using this notation we rewrite the bound in (78) as

$$
\|\Sigma_\xi' - \Sigma_\eta'\|_{\mathrm{F}} \le (\varepsilon C_{\Sigma,1}(d) + \varepsilon^2 C_{\Sigma,2}(d))W_2(\xi,\eta) \,.
\tag{79}
$$

By definition of the Bures distance in (Bhatia et al., 2019)[Theorem 1],

$$
d_B(\Sigma_\xi', \Sigma_\eta') \le \|(\Sigma_\xi')^{1/2} - (\Sigma_\eta')^{1/2}\|_{\mathrm{F}} \,,
\tag{80}
$$

and by (Bhatia, 2013)[Theorem X.3.8], we have

$$
\|(\Sigma_\xi')^{1/2} - (\Sigma_\eta')^{1/2}\|_{\mathrm{F}} \le \frac{1}{2\sqrt{\lambda_{\min}'}}\|\Sigma_\xi' - \Sigma_\eta'\|_{\mathrm{F}} \,,
\tag{81}
$$

where $\lambda_{\min}'$ is the lower bound on the eigenvalues of $\Sigma_\xi', \Sigma_\eta'$. In the next few steps of the proof we show that $\lambda_{\min}' \ge \frac{\lambda_{\min}}{2}$. By (Bhatia, 2013)[Corollary III.2.6], we have that for Hermitian matrices $A, B$:

$$
\max_j |\lambda_j(A) - \lambda_j(B)| \le \|A - B\|_{\mathrm{op}} \,.
$$

Since $\Sigma_\xi' = \tilde{A}\Sigma_\xi\tilde{A}^T$ and $\Sigma_i$ are symmetric, from the above inequality, we obtain

$$
|\lambda_{\min}(\Sigma_\xi') - \lambda_{\min}(\Sigma_i)| \le \|\Sigma_\xi' - \Sigma_i\|_{\mathrm{op}} \,,
$$

and therefore, $\lambda_{\min}' \ge \lambda_{\min} - \|\Sigma_\xi' - \Sigma_i\|_{\mathrm{op}}$. Next, the bound in (22) yields

$$
\lambda_{\min}' \ge \lambda_{\min} - \lambda_{\max}\varepsilon(3\kappa^2 + 2) = \lambda_{\min}\left(1 - \kappa\varepsilon(3\kappa^2 + 2)\right) \,.
\tag{82}
$$

Now, the constraint $\beta \ge \max\{1, \beta_0^{(2)}\}$ in the statement of the lemma implies $\kappa\varepsilon(3\kappa^2 + 2) \le 1/2$ for $d > \frac{(3\kappa^2+2)^2}{288e^4\kappa^{10}}$. By plugging this into (82), we get $\lambda_{\min}' \ge \lambda_{\min}/2$, which when applied to the inequality in (81) yields

$$
\|(\Sigma_\xi')^{1/2} - (\Sigma_\eta')^{1/2}\|_{\mathrm{F}} \le \frac{1}{\sqrt{2\lambda_{\min}}}\|\Sigma_\xi' - \Sigma_\eta'\|_{\mathrm{F}} \,.
\tag{83}
$$

Hence by using the bounds in (79) and (80), we get

$$d_B(\Sigma'_\xi, \Sigma'_\eta) \leq \frac{1}{\sqrt{2\lambda_{\min}}}(\varepsilon C_{\Sigma,1}(d) + \varepsilon^2 C_{\Sigma,2}(d))W_2(\xi, \eta). \tag{84}$$

By plugging the bounds in (63) and (84) into (40), using $\sqrt{a^2 + b^2} \leq |a| + |b|$, and using $\varepsilon < 1$, we get

$$\begin{aligned}
W_2(\Phi(\xi), \Phi(\eta)) &\leq \left[\varepsilon C_m + \frac{(\varepsilon C_{\Sigma,1}(d) + \varepsilon^2 C_{\Sigma,2}(d))}{\sqrt{2\lambda_{\min}}}\right] W_2(\xi, \eta) \\
&\leq \varepsilon \left[C_m(d) + \frac{C_{\Sigma,1}(d) + C_{\Sigma,2}(d)}{\sqrt{2\lambda_{\min}}}\right] W_2(\xi, \eta) \\
&= L W_2(\xi, \eta),
\end{aligned} \tag{85}$$

where $C_m(d) := 4\beta M_W(4M_W + C_1(d))$ and $L := \varepsilon\left[C_m(d) + \frac{C_{\Sigma,1}(d) + C_{\Sigma,2}(d)}{\sqrt{2\lambda_{\min}}}\right]$. We introduce more notation to simplify the next steps:

$$\eta_1(d) := 4M_W(4M_W + C_1(d)) + \frac{24\kappa\lambda_{\max}\sqrt{d}(4M_W + C_1(d))}{\sqrt{2\lambda_{\min}}}, \ \eta_2(d) := \frac{1}{\sqrt{2\lambda_{\min}}}\left(\frac{48\sqrt{d}\kappa\lambda_{\max}^5}{\lambda_{\min}^{9/2}} + \frac{48\kappa^2\lambda_{\max}}{\sqrt{\lambda_{\min}}}\right).$$

With this notation note that $C_m(d) + C_{\Sigma,1}(d) + C_{\Sigma,2}(d) = \beta\eta_1(d) + \eta_2(d)$. For $\beta \geq 1$, we have $\beta\eta_1(d) + \eta_2(d) < \beta(\eta_1(d) + \eta_2(d))$. This means $L < \varepsilon\beta(\eta_1(d) + \eta_2(d))$. Finally, we show that $\varepsilon\beta(\eta_1(d) + \eta_2(d)) < 1$ which would imply $L < 1$. In the statement of the lemma, we assumed $\beta \geq \beta_0 = \frac{\log N + \log(\eta_1(d) + \eta_2(d)) + 2}{2\lambda_{\min}d - 1}$. For $\beta \geq \beta_0$, this constraint on $\beta$ implies $2\beta\lambda_{\min}d - \beta \geq \log N(\eta_1(d) + \eta_2(d)) + 2$. Next, $-\log\beta \geq -\beta$ implies $2\beta\lambda_{\min}d - \log\beta \geq \log N(\eta_1(d) + \eta_2(d)) + 2$. Taking exponentials and rearranging terms of this inequality, we get $Ne^{-2\beta\lambda_{\min}d}\beta(\eta_1(d) + \eta_2(d)) \leq e^{-2}$. Since $e^{-2} < 1$ we have $Ne^{-2\beta\lambda_{\min}d}\beta(\eta_1(d) + \eta_2(d)) < 1$. Since $\varepsilon = Ne^{-2\beta\lambda_{\min}d}$, we have shown that $\varepsilon\beta(\eta_1(d) + \eta_2(d)) < 1$, and therefore $L < \varepsilon\beta(\eta_1(d) + \eta_2(d)) < 1$. This finishes the proof.

$\square$

**Remark 4.** *Lemma 3 proves that the map $\Phi$ is a self-map and Lemma 4 proves that $\Phi$ is a contractive map. The following lemma puts these two facts together to prove the existence of a unique fixed point in a neighborhood of the Gaussian measure $X_i$ for all $i \in [N]$.*

**Lemma 5.** *Let $\{X_i = \mathcal{N}(\mu_i, \Sigma_i)\}_{i=1}^N$ be N d-dimensional Gaussian measures such that $\Sigma_i \succ 0$ and the eigenvalues of $\Sigma_i$ lie in the bounded interval $[\lambda_{\min}, \lambda_{\max}]$ for all $i \in [N]$. Define the Wasserstein balls $\mathcal{B}_i$ around the Gaussian measure $X_i$, for $i \in [N]$, as $\mathcal{B}_i := \{\nu \in \mathcal{P}_2(\mathbb{R}^d) : W_2(X_i, \nu) \leq \sqrt{\lambda_{\min}}\}$. Let $\beta_0^{(1)}, \beta_0^{(2)}$ be as defined in Lemma 3 and Lemma 4 respectively. Assume the separation condition*

$$\Delta_i := \min_{i \neq j}(-\log\langle X_i, X_j\rangle_{L^2}) \geq \frac{d}{2}\log(4\pi\lambda_{\max}) + d.$$

*If $d \geq 36$, $2\lambda_{\min}d > 1$, and $\beta \geq \max\{1, \beta_0^{(1)}, \beta_0^{(2)}\}$, then the map $\Phi$ has a unique fixed point in $\mathcal{B}_i$ for all $i \in [N]$.*

*Proof.* Since $\mathcal{B}_i$ is a closed subset of $\mathcal{P}_2(\mathbb{R}^d)$ and since $(\mathcal{P}_2(\mathbb{R}^d), W_2)$ is a complete metric space, we have that $(\mathcal{B}_i, W_2)$ is a complete metric space. Finally, since for $2\lambda_{\min}d > 1$ and $\beta \geq \max\{1, \beta_0^{(1)}, \beta_0^{(2)}\}$, we have that $\Phi$ is a self-map by Lemma 3 and a contractive map by Lemma 4, we conclude that $\Phi$ has a unique fixed point in $\mathcal{B}_i$, for $i \in [N]$, by Banach's fixed point theorem. $\square$

The next steps are to prove storage capacity of the energy functional defined in (1). We begin by providing an algorithm to sample Gaussian measures from a Wasserstein sphere.

---

**Algorithm 2** Sampling Gaussian Measures from a Wasserstein Sphere

---

**Require:** Dimension $d$, spectral bounds $0 < \lambda_{\min} \leq \lambda_{\max} < \infty$, number of samples $N$
**Ensure:** Set of Gaussian measures $\{X_i = \mathcal{N}(\mu_i, \Sigma_i)\}_{i=1}^N$ on a Wasserstein sphere of radius $R$ centered at $\delta_0$
  1: **for** $i = 1, \ldots, N$ **do**
  2:    **Sample eigenvalues:** Generate $\lambda_1^{(i)}, \ldots, \lambda_d^{(i)} \overset{\text{i.i.d.}}{\sim} \text{Uniform}([\lambda_{\min}, \lambda_{\max}])$
  3:    Compute trace $\tau_i \leftarrow \sum_{k=1}^d \lambda_k^{(i)}$
  4:    **Sample orthogonal matrix from the Haar measure:**
  5:    Generate $Z \in \mathbb{R}^{d \times d}$ with entries $Z_{jk} \overset{\text{i.i.d.}}{\sim} \mathcal{N}(0, 1)$
  6:    Compute $QR$ decomposition of $Z$, i.e., $Z = QR$
  7:    $Q_i \leftarrow Q \cdot \text{diag}(\text{sign}(R_{11}), \ldots, \text{sign}(R_{dd}))$
  8:    **Construct covariance matrix:** $\Sigma_i \leftarrow Q_i \, \text{diag}(\lambda_1^{(i)}, \ldots, \lambda_d^{(i)}) \, Q_i^\top$
  9:    **Sample mean:**
 10:    Sample $Z \sim \mathcal{N}(0, I_d)$
 11:    $\mu_i \leftarrow \sqrt{R^2 - \tau_i} \cdot \frac{Z}{\|Z\|}$
 12:    $X_i \leftarrow \mathcal{N}(\mu_i, \Sigma_i)$
 13: **end for**
 14: Return the Gaussian measures $\{X_i\}_{i=1}^N$

---

**Remark 5.** *Note that $QR$ decomposition of the real matrix $Z$ in Algorithm 2 implies $Q$ is orthogonal. The proof of the claim in Algorithm 2 that the matrices $Q_i$, defined in line 7 of Algorithm 2, are distributed according to the Haar measure on the space of orthogonal matrices $O(d)$ is provided in Mezzadri (2006)[Theorem 1].*

*Proof of Theorem 1.* Let $X_i = \mathcal{N}(\mu_i, \Sigma_i)$ and $X_j = \mathcal{N}(\mu_j, \Sigma_j)$. By definition of the $L^2$-inner product between Gaussian measures, we have

$$-\log\langle X_i, X_j \rangle_{L^2} = \frac{d}{2}\log(2\pi) + \frac{1}{2}\log|\Sigma_i + \Sigma_j| + \frac{1}{2}(\mu_i - \mu_j)^\top (\Sigma_i + \Sigma_j)^{-1}(\mu_i - \mu_j). \tag{86}$$

By construction in Algorithm 2, the eigenvalues of $\Sigma_i$, for all $i \in [N]$, lie in the bounded interval $[\lambda_{\min}, \lambda_{\max}]$ almost surely. Therefore, for any pair $i, j$, the eigenvalues of the sum $\Sigma_i + \Sigma_j$ lie in $[2\lambda_{\min}, 2\lambda_{\max}]$ almost surely, and we obtain $|\Sigma_i + \Sigma_j| \geq (2\lambda_{\min})^d$ and $(\Sigma_i + \Sigma_j)^{-1} \succeq (2\lambda_{\max})^{-1}I$ almost surely. Applying these to (86), we get, almost surely

$$-\log\langle X_i, X_j \rangle_{L^2} \geq \frac{d}{2}\log(2\pi) + \frac{d}{2}\log(2\lambda_{\min}) + \frac{\|\mu_i - \mu_j\|^2}{4\lambda_{\max}}.$$

To establish the desired separation condition in the statement of this theorem, it suffices to show that for all $i \neq j$:

$$\frac{d}{2}\log(4\pi\lambda_{\min}) + \frac{\|\mu_i - \mu_j\|^2}{4\lambda_{\max}} \geq \frac{d}{2}\log(4\pi\lambda_{\max}) + d.$$

Rearranging this inequality and using $\kappa = \lambda_{\max}/\lambda_{\min}$, we get

$$\|\mu_i - \mu_j\|^2 \geq 2d\lambda_{\max}(\log\kappa + 2). \tag{87}$$

By Algorithm 2, we have that for all $i$: $\mu_i = r_i u_i$ where $u_i$ is uniform on the unit sphere $\mathbb{S}^{d-1}$ and $r_i = \sqrt{R^2 - \tau_i}$ with $\tau_i = \text{tr}(\Sigma_i)$. Since each eigenvalue lies in the interval $[\lambda_{\min}, \lambda_{\max}]$, the trace satisfies $\tau_i \in [d\lambda_{\min}, d\lambda_{\max}]$ almost surely. Using $R^2 = 2d\lambda_{\max}(2 + \log\kappa)$, we get the following bound on $r_i$:

$$r_i^2 = R^2 - \tau_i \geq 2d\lambda_{\max}(2 + \log\kappa) - d\lambda_{\max} = d\lambda_{\max}(3 + 2\log\kappa) \text{ a.s.}.$$

Define $r_{\min} := \sqrt{d\lambda_{\max}(3 + 2\log\kappa)}$. Next, We claim that $\|\mu_i - \mu_j\|^2 \geq 2r_{\min}^2(1 - \langle u_i, u_j \rangle)$ almost surely. To verify this, fix $t \leq 1$ and consider the function $f(r_i, r_j) = r_i^2 + r_j^2 - 2r_i r_j t$ over the region $\mathcal{D} := [r_{\min}, \infty) \times [r_{\min}, \infty)$. We will show that $f(r_i, r_j) \geq 2r_{\min}^2(1 - t)$ for all $(r_i, r_j) \in [r_{\min}, \infty) \times [r_{\min}, \infty)$. For $t = 1$, $f(r_i, r_j) = (r_i - r_j)^2 \geq 0 = 2r_{\min}^2(1 - t)$. For $t < 1$, the Hessian of $f$ is $\begin{bmatrix} 2 & -2t \\ -2t & 2 \end{bmatrix}$, which has eigenvalues $2(1 - t) > 0$ and $2(1 + t) > 0$. Thus $f$ is strictly

convex. The unique critical point, found by setting $\nabla f = 0$ is $(r_i, r_j) = (0, 0)$ lies outside the domain of $f$. Since $f$ is strictly convex and $\mathcal{D}$ is a convex set, the minimum of $f$ is attained at $(r_{\min}, r_{\min})$ and hence $f(r_i, r_j) \geq 2r_{\min}^2 (1-t)$ for all $t \leq 1$. By noting $\langle u_i, u_j \rangle \leq \|u_i\| \cdot \|u_j\| = 1$ and $\|\mu_i - \mu_j\|^2 = r_i^2 + r_j^2 - 2\langle u_i, u_j \rangle$, and plugging $t = \langle u_i, u_j \rangle$ in the function $f$, we get the claim: $\|\mu_i - \mu_j\|^2 \geq 2r_{\min}^2(1 - \langle u_i, u_j \rangle)$ almost surely.

From (87) it follows that it suffices to prove $2r_{\min}^2(1 - \langle u_i, u_j \rangle) \geq 2d\lambda_{\max}(\log \kappa + 2)$, for all $i \neq j$, to establish that the desired separation condition in the statement of this theorem holds. This inequality is equivalent to

$$\langle u_i, u_j \rangle \leq 1 - \frac{2 + \log \kappa}{3 + 2\log \kappa}.$$

To simplify notation in the next steps, we define $\delta := \frac{2+\log \kappa}{3+2\log \kappa}$.

By Vershynin (2009)[Theorem 3.4.5], we have that for independent random vectors $u, v$ on a unit sphere $\mathbb{S}^{d-1}$ and $t > 0$:

$$\mathbb{P}(\langle u, v \rangle > t) \leq 2\exp(-t^2 d/2). \tag{88}$$

Since $u_i, u_j$ are independent random vectors by Algorithm 2, by (88), we get

$$\mathbb{P}(\langle u_i, u_j \rangle \geq 1 - \delta) \leq 2\exp(-(1-\delta)^2 d/2).$$

Since there are $\binom{N}{2} < \frac{N^2}{2}$ pairs of distinct indices $i, j$, by the union bound, we obtain

$$\mathbb{P}(\exists i \neq j \, : \, \langle u_i, u_j \rangle \geq 1 - \delta) \leq N^2 \exp(-(1-\delta)^2 d/2). \tag{89}$$

With $N = \lfloor \sqrt{p} \exp \left( (1-\delta)^2 d/4 \right) \rfloor$ we have $N^2 \leq p \cdot \exp \left( 2(1-\delta)^2 d/4 \right) = p \cdot \exp \left( d(1-\delta)^2/2 \right)$. By plugging this bound on $N^2$ in (89), we get

$$\mathbb{P}(\exists i \neq j \, : \, \langle u_i, u_j \rangle \geq 1 - \delta) \leq p \exp \left( d(1-\delta)^2/2 \right) \cdot \exp(-(1-\delta)^2 d/2) = p.$$

The above inequality proves that with probability at least $1 - p$, all pairs of sampled Gaussian measures using Algorithm 2 satisfy the separation condition defined in the statement of this theorem.

Next, by using the definition of $M_W$ and by observing that all the sampled Gaussian measures are on the Wasserstein sphere of radius $R$, we obtain

$$M_W = \max_{i \in [N]} W_2(\delta_0, X_i) = R = \sqrt{2d\lambda_{\max}(2 + \log \kappa)}, \tag{90}$$

The final steps of the proof are aimed toward showing $\beta_0^{(1)}$, as defined in Lemma 3, and $\beta_0^{(2)}$, as defined in Lemma 4, satisfy $\beta_0^{(1)} < \beta_0^{(2)}$ for all $d \geq d_0$ and $M_W = \sqrt{2d\lambda_{\max}(2 + \log \kappa)}$, where $d_0 := \lfloor \max \left\{ 36, \frac{1}{2\lambda_{\min}}, \frac{(c_1+c_2)^2}{e^4 c_3^2}, \frac{\log(1/p)}{2\alpha} \right\} \rfloor + 1$ and

$$c_1 := 16\kappa(2 + \log \kappa), \, c_2 := 2\kappa(3\kappa^2 + 2), \, c_3 := \frac{192\kappa^2 \lambda_{\max}}{\sqrt{2\lambda_{\min}}} \sqrt{2\lambda_{\max}(2 + \log \kappa)}.$$

To simplify notation in the next steps, define

$$A_1(d) := \frac{8M_W^2}{\lambda_{\min}} + 2d\kappa(3\kappa^2 + 2),$$

$$A_2(d) := \left( 4M_W + \frac{24\kappa\lambda_{\max}\sqrt{d}}{\sqrt{2\lambda_{\min}}} \right) \left( 4M_W + 4\kappa(3 + \sqrt{d}) \left( \sqrt{\lambda_{\min}} + 2M_W \right) \right) + \frac{1}{\sqrt{2\lambda_{\min}}} \left( \frac{48\sqrt{d}\kappa\lambda_{\max}^5}{\lambda_{\min}^{9/2}} + \frac{48\kappa^2\lambda_{\max}}{\sqrt{\lambda_{\min}}} \right).$$

Since $M_W = \sqrt{2d\lambda_{\max}(2 + \log \kappa)}$ from (90), we have

$$A_1(d) = (c_1 + c_2)d. \tag{91}$$

Since all the terms in $A_2(d)$ are positive, using $M_W = c\sqrt{d}$, and using the notation $\eta_1(d), \eta_2(d)$ from Lemma 4, we get

$$A_2(d) = \eta_1(d) + \eta_2(d) \geq \eta_1(d) \geq (8\kappa cd) \cdot \frac{24\kappa\lambda_{\max}\sqrt{d}}{\sqrt{2\lambda_{\min}}} \geq c_3 d^{3/2}, \tag{92}$$

where $c_3 := \frac{192\kappa^2 c\lambda_{\max}}{\sqrt{2}\lambda_{\min}}$. Combining (91) and (92), we get

$$\frac{A_2(d)}{A_1(d)} \geq \frac{c_3}{c_1 + c_2}\sqrt{d}\,. \tag{93}$$

Using $d \geq d_0 \geq \frac{(c_1+c_2)^2}{e^4 c_3^2}$ in (93), we get

$$\log\left(\frac{A_2(d)}{A_1(d)}\right) + 2 > 0\,. \tag{94}$$

Now, by definition of $\beta_0^{(1)}$ in Lemma 3 and $\beta_0^{(2)}$ in Lemma 4, the condition $\beta_0^{(1)} < \beta_0^{(2)}$ is equivalent to

$$\frac{\log N + \log A_1(d)}{2\lambda_{\min}d} < \frac{\log N + \log A_2(d) + 2}{2\lambda_{\min} - 1}\,.$$

Rearranging the above terms, we get

$$-(\log N + \log A_1(d)) < 2\lambda_{\min}d\left(\log\left(\frac{A_2(d)}{A_1(d)}\right) + 2\right)\,. \tag{95}$$

Note that $N = \sqrt{p}e^{\alpha d} > 1$ for $d \geq d_0 > \frac{\log(1/p)}{2\alpha}$. Also, by definition $A_1(d) > 1$. By using these two facts, we have that $-(\log N + \log A_1(d)) < 0$, and by using the bound in (94), we have $2\lambda_{\min}d\left(\log\left(\frac{A_2(d)}{A_1(d)}\right) + 2\right) > 0$. Therefore, (95) holds for all $d \geq d_0$ and consequently, $\beta_0^{(1)} < \beta_0^{(2)}$ for all $d \geq d_0$.

Finally, by noting that the value of $\beta_0$ in the statement of this theorem is equal to the value of $\beta_0^{(2)}$ in Lemma 4 with $N = \lfloor\sqrt{p}e^{\alpha d}\rfloor$ and $M_W = \sqrt{2d\lambda_{\max}(2 + \log\kappa)}$, by noting that the separation condition in the statement of this theorem is satisfied with probability $1 - p$, and by using Lemma 5, we get that for all $\beta \geq \beta_0$, there exists a unique fixed point in a neighborhood of $X_i$. Moreover, since $W_2(X_i, X_j) > 2\sqrt{\lambda_{\min}d} \geq 2\sqrt{\lambda_{\min}}$, for $d \geq 1$, from (34), the Wasserstein balls $\mathcal{B}_i$ are pairwise disjoint, satisfying the second condition in Definition 1. Consequently, by the definition of storage of a Gaussian measure in Definition 1, we can store $N = \lfloor\sqrt{p}e^{\alpha d}\rfloor$ Gaussian measures with probability $1 - p$ for all $\beta \geq \beta_0$. $\qquad\square$

## 6.4. Proof of Theorem 2

*Proof of Theorem 2.* Part (1) follows from Lemma 3, which shows that $\Phi$ maps $\mathcal{B}_i$ to itself under the assumptions stated in this theorem.

For part (2), Lemma 4 establishes that $\Phi$ is a contraction on $\mathcal{B}_i$ with constant $L < 1$. By Banach's fixed point theorem, there exists a unique fixed point $X_i^* \in \mathcal{B}_i$. Since $X_i^* = \Phi(X_i^*)$, we have

$$W_2(\xi^{(k+1)}, X_i^*) = W_2(\Phi(\xi^{(k)}), \Phi(X_i^*)) \leq L \cdot W_2(\xi^{(k)}, X_i^*)\,.$$

Applying this inequality inductively yields

$$W_2(\xi^{(k)}, X_i^*) \leq L^k \cdot W_2(\xi^{(0)}, X_i^*)\,.$$

The bound $W_2(\xi^{(0)}, X_i^*) \leq 2\sqrt{\lambda_{\min}}$ follows from the triangle inequality: since $\xi^{(0)}, X_i^* \in \mathcal{B}_i$, we have $W_2(\xi^{(0)}, X_i^*) \leq W_2(\xi^{(0)}, X_i) + W_2(X_i, X_i^*) \leq 2\sqrt{\lambda_{\min}}$.

Part (3) follows by solving $2\sqrt{\lambda_{\min}} \cdot L^k \leq \varepsilon$ for $k$, which gives $k \geq \log(2r/\varepsilon)/\log(1/L)$. $\qquad\square$

## 6.5. Proof of Theorem 3

*Proof.* From (26) in the proof of Lemma 3:

$$W_2^2(\Phi(\xi^{(0)}), X_i) \leq 4\varepsilon M_W^2 + d(3\kappa^2 + 2)\varepsilon\lambda_{\max}\,,$$

where $\varepsilon = Ne^{-2\beta\lambda_{\min}d}$. Since the Gaussian measures are sampled from a Wasserstein sphere of radius $R = \sqrt{2d\lambda_{\max}(2 + \log\kappa)}$, we have $M_W = R$. Plugging this into the above equation yields

$$W_2^2(\Phi(\xi^{(0)}), X_i) \leq \varepsilon d\lambda_{\max} \cdot \left[8(2 + \log\kappa) + (3\kappa^2 + 2)\right]\,.$$

Since $N = \lfloor \sqrt{p} e^{\alpha d} \rfloor \leq \sqrt{p} e^{\alpha d}$, we have

$$\varepsilon = N e^{-2\beta\lambda_{\min} d} \leq \sqrt{p} e^{(\alpha - 2\beta\lambda_{\min})d} \, .$$

Plugging the above bound into the previous inequality yields

$$W_2^2(\Phi(\xi^{(0)}), X_i) \leq \sqrt{p} e^{(\alpha - 2\beta\lambda_{\min})d} d\lambda_{\mathsf{max}} \cdot \left[ 8(2 + \log\kappa) + (3\kappa^2 + 2) \right] \, .$$

Finally, taking square roots of both the sides of the above inequality and using $\beta \geq \frac{\alpha}{2\lambda_{\min}}$ proves this theorem. $\qquad\square$

