# OpenReview forum: "Dense associative memory for Gaussian distributions"
_ICML.cc/2026/Conference — ICML 2026 regular_

### Official Review · Reviewer_ywbS · 2026-03-06

**Soundness:** 3
**Presentation:** 4
**Significance:** 3
**Originality:** 3
**Overall Recommendation:** 4
**Confidence:** 4

**Summary:**

This paper generalizes dense associative memory to the 2-Wasserstein distance. This allows DAM to store and recall probability distributions. Specifically, this paper focuses on Gaussian distribution.

**Compliance With Llm Reviewing Policy:**

Affirmed.

**Key Questions For Authors:**

1. The comparison in remark 2 does not seem fair to me.  In remark 2, the basin sizes are set to be $\sqrt{\lambda_{\min}}$. The authors claim that this is an improvement comparing to DAM or MHN. However, from what I understand about Gaussian measures, the separation condition (L2 distance between Gaussian measures) depends on $\lambda_{\min}$. From a packing perspective, $\lambda_{\min}$ directly controls the maximum possible value of $N$. This makes the basin sizes depends on the minimal separation between measures and further depends on $N$. Therefore, to me the trade-off still exists.

2. Is the basin size dependent on $\beta$? It seems counter-intuitive if it is independent to the temperature.

3. What is the difficulty of generalizing the result to other distributions? Is it mostly about the closed-form W_2 distance?

**Limitations:**

I did not find the limitation paragraph or section.

**Strengths And Weaknesses:**

### Strengths

1. This paper is very well-written and easy to follow, especially for readers familiar with Hopfield networks.
2. While I did not check every single statistical rates in the paper, the results make sense to me.
3. The results in this paper is novel to my knowledge. It is the first paper that extends associative memory models to $\mathcal{P}_2(\mathbb{R}^d)$.
4. The work is significant as it (a) proposes a novel associative memory model with non-trivial modifications to DAM, (b) it provides a new perspective to understand diffusion models.

### Weaknesses

1. The proofs are difficult to find for each theorem. I suggest the authors to add links to the proofs.
2. The proposed update rule is not trivial computationally comparing to closed form updates in DAM, especially when we only observe samples from each stored Gaussian measure. I think adding discussions or sample complexities to this case would be useful in practice. It is also worth discussing the time complexity of algorithm 1.

---

> ### Author Rebuttal · Authors · 2026-03-28
>
> 1. First, we thank the reviewer for identifying the strengths and originality of our paper.
>
> 2. We shall add links to the proofs of all the theorems in the main section of the paper in the revision.
>
> 3. Each iteration of Algorithm 1 is dominated by $N$ eigen-decompositions of $d\times d$ matrices, since for each stored Gaussian measure $X_i$, computing $D_i = W_2^2(X_i, \xi)$ and the transport map coefficient $A_i$ requires forming the product $\Sigma_i^{1/2} \Omega \Sigma_i^{1/2}$ via two $d\times d$ matrix multiplications, each costing $O(d^3)$ by standard matrix multiplication, and then eigen-decomposing the resulting $d\times d$ positive symmetric definite matrix also costs $O(d^3)$ via, e.g., the QR algorithm, while the softmax weights $w_i$, weighted mean $m'$, weighted transport map $\tilde{A}$ and the covariance update contribute $O(Nd^2 + d^3)$. The per-iteration complexity of Algorithm 1 is therefore $O(Nd^3)$.
>
>    We agree that the paper would benefit from addition of details about complexity of Algorithm 1. We shall add it to the revision.
>
> 4. Your packing argument: $N$ depends on $\lambda_{\min}$ is correct. However, we consider the case where $\lambda_{\min}, \lambda_{\max}$ are fixed. In this case, $N$ grows exponentially in $d$ while the BW-DAM basin radius remains fixed at $\sqrt{\lambda_{\min}}$.
>
> 5. The basin size used in the lemmas in the paper is indeed independent of $\beta$. Note that in Lemma 4 we prove one-step contraction in $W_2$ distance for $\beta \geq \beta_0$. Hence, the idea is that for *large enough $\beta$*, the basin radius $r = \sqrt{\lambda_{\min}}$ is sufficient for contraction. Although the basin radius $r = \sqrt{\lambda_{\min}}$ is sufficient in Lemma 4, we conjecture that the basin radius grows with $\beta$, because as $\beta \to \infty$, the weights concentrate almost entirely on the nearest Gaussian. On the other hand, we believe it is counter-intuitive that in [Ramsauer et al., 2020], the proposed basin radius $r = 1/\sqrt{\beta N}$, is inversely proportional to $\beta$. In other words, we believe the basin radius used for contraction in [Ramsauer et al., 2020] is much smaller than what is possible.

---

> > ### Author Rebuttal · Reviewer_ywbS · 2026-04-03
> >
> > My concerns are fully resolved.

---

### Official Review · Reviewer_TbdR · 2026-03-11

**Soundness:** 3
**Presentation:** 3
**Significance:** 4
**Originality:** 4
**Overall Recommendation:** 5
**Confidence:** 4

**Summary:**

This paper extends the classical Dense Associative Memory (DAM) framework from vectors to probability distributions, specifically focusing on Gaussian distributions. The authors define an energy functional in the Wasserstein space using the log-sum-exp of squared Wasserstein-2 distances, and derive the corresponding gradient flow and fixed-point conditions. They specialize to the Bures-Wasserstein manifold of Gaussian distributions, obtaining closed-form expressions for the retrieval dynamics (Algorithm 1). The main theoretical contributions are: (i) a proof that exponentially many (in dimension $d$) Gaussian measures can be stored as attractors under suitable separation conditions and sufficiently large inverse temperature $\beta$ (Theorem 1); (ii) contraction guarantees ensuring convergence to unique fixed points within basins of attraction (Theorem 2); (iii) a bound on the one-step retrieval error that decays exponentially in $d$ (Theorem 3). Experiments on synthetic data, CelebA, CIFAR-10, and sentence embeddings demonstrate the effectiveness of the proposed Bures-Wasserstein DAM (BW-DAM) compared to Euclidean baselines.

**Compliance With Llm Reviewing Policy:**

Affirmed.

**Key Questions For Authors:**

The following questions will directly impact my evaluation of the paper, and may lead to an upgrade in my ratings and overall recommendation if fully addressed.
1. Can the framework be extended beyond Gaussian distributions?
2. Poor readability: (1) ``Notation and definitions.'', this part is is very crowded.
		(2) The use of abbreviations is highly irregular. Please use the full name (abbreviation) the first time it appears, and use the abbreviation thereafter. e.g. Dense associative memory (DAM); Bures–Wasserstein (BW); Log-Sum-Exp
		(LSE); Euclidean DAM (Eu-DAM); Bures–Wasserstein DAM (BW-DAM);
		(3) Typographical errors: There are many typos. e.g., ``(Bures)-Wasserstein space'' in line 58-59 left; ``extending extending'' in line 117-118 left;
		(4) The use of brackets in formulas is very limited; please use a combination of parentheses, square brackets, and curly brackets. In line 275-276, ``new'' in equation should be non-italic.
3. The citation format for references is inconsistent. (1) Only four references provide URL links, I think you should remove them to keep consistency. (2) Please maintain consistency in your references. For example, ``In Forty-second International Conference on Machine Learning, 2025.'', and ``In International Conference on Machine Learning (ICML),''; ``In Advances in Neural Information Processing Systems,'' , ``In The Thirty-eighth Annual Conference on Neural Information Processing Systems'', and ``Advances in Neural Information Processing Systems,''; etc.

**Limitations:**

yes

**Strengths And Weaknesses:**

The paper presents a novel extension of dense associative memory to the space of Gaussian distributions by formulating the energy function and retrieval dynamics within the Bures–Wasserstein metric. This geometric adaptation preserves the structure of probabilistic representations and enables associative memory to operate directly on distributions rather than vectors. The theoretical analysis is substantial, providing rigorous proofs of exponential storage capacity under precise separation conditions, and the retrieval algorithm is clearly specified with closed-form updates for both mean and covariance. The model is evaluated across multiple domains—including synthetic data, image reconstruction with VAEs, and sentence embeddings—demonstrating consistent improvements over Euclidean baselines and validating the advantages of respecting Wasserstein geometry in distributional memory tasks.

	Despite its theoretical contributions, the paper insufficiently clarifies its novelty relative to existing work on distributional representations and energy-based models, lacking a systematic comparison with recent memory-augmented architectures. The core theoretical guarantees rely on strong assumptions—such as bounded covariance eigenvalues and explicit $L^2$ separation—that are explicitly not verified in the real-data experiments, creating a disconnect between the analytical claims and empirical validation. The experimental evaluation is limited to comparisons with Euclidean and pixel-space DAMs, omitting more relevant baselines like modern Hopfield networks or energy transformers. Furthermore, the retrieval procedure requires computing Wasserstein distances to all stored memories, incurring linear computational cost in the number of patterns, which may limit scalability to large memory stores without further efficiency considerations.

---

> ### Author Rebuttal · Authors · 2026-03-28
>
> 1. First, we thank the reviewer for identifying the strengths and originality of our paper.
>
> 2. It is true that our paper only addresses the case of Gaussian distributions. The main challenges with extending it to arbitrary probability distributions are:
>
>    - The 2-Wasserstein distance between two arbitrary probability distributions need not have a closed-form expression.
>
>    - The optimal transport map from a probability distribution $X_1$ to $X_2$ need not be affine. In our paper, the optimal transport map $T$ from $X_1 = \mathcal{N}(\mu_1, \Sigma_1)$ to $X_2 = \mathcal{N}(\mu_2, \Sigma_2)$ is affine:
>      $$T(x) = \mu_2 + A(x - \mu_1) $$
>
>    - Moreover, our paper characterized the separation between Gaussian distributions in $L^2$ distance which has a clean closed-form expression involving the means and covariance matrices. This need not exist for arbitrary probability distributions.
>
>    - The main theoretical tool we used in the paper is Banach's fixed point theorem to guarantee the existence of a unique fixed point in the neighborhood of a Gaussian distribution. The above three facts make proving one-step contraction in $W_2$ distance, which we did for the Gaussian case in the paper in Lemma 4 in the paper, quite challenging.
>
> 3. We shall fix both the readability issues and reference formatting issues you raised in the revision.
>
> 4. In the paper we made empirical comparisons with Dense Associative Memory which is a modern Hopfield network, because we thought that is an appropriate comparison. However, we realize the importance of transformer style memory mechanisms. Hence, in our revision, we will add a comparison to energy transformers.

---

> > ### Author Rebuttal · Reviewer_TbdR · 2026-04-01
> >
> > I believe the current score is appropriate for this article and there is no need to raise it further. I am very grateful to the author for their contribution to this article.

---

### Official Review · Reviewer_fasU · 2026-03-11

**Soundness:** 3
**Presentation:** 3
**Significance:** 3
**Originality:** 3
**Overall Recommendation:** 4
**Confidence:** 2

**Summary:**

The paper studies how dense associative memory models can be extended to operate on probability distributions rather than deterministic vectors. The authors propose a Wasserstein-based DAM that measures similarity between stored Gaussian distributions and query distributions using the 2-Wasserstein distance and defines a log-sum-exp energy over these distances. Retrieval is performed through weighted optimal transport updates that iteratively move the query distribution toward a self-consistent Wasserstein barycenter of nearby stored memories. The paper provides theoretical guarantees: the proposed model preserves desirable properties of dense associative memories, including exponential storage capacity and convergence of retrieval when queries lie within a Wasserstein neighborhood of stored patterns. Experiments on synthetic data and real Gaussian embeddings, including CelebA image embeddings and Word2Gauss text embeddings, demonstrate that the proposed framework that respects the underlying distributional geometry improves retrieval accuracy compared with Euclidean baselines that treat Gaussian parameters as vectors.

**Compliance With Llm Reviewing Policy:**

Affirmed.

**Final Justification:**

This is an interesting work and I remain positive about it. Due to the potential of the  proposed framework is not yet fully explored, I will maintain my current score and not raise it further.

**Key Questions For Authors:**

Could the authors comment on whether the proposed Wasserstein DAM framework is expected to scale to more complex visual datasets or higher-dimensional embeddings, such as those arising from large-scale image models?

**Limitations:**

Yes

**Strengths And Weaknesses:**

**Strengths**:
* The extension of DAM to probability distribution is an interesting route to explore.
* The work provides rigorous results including exponential storage capacity and convergence guarantees in Wasserstein space.
* Experimental part covers both image and test domains.

**Weaknesses**:
* The framework only handles Gaussian distributions with closed-form Wasserstein distances, thus limiting the practial usage of the proposed framework.
* The paper mainly compares against Euclidean DAM baselines but does not evaluate against more recent and powerful attention-based or transformer-style memory mechanisms.

Minor:
Line 118 "This motivates extending extending..."

---

> ### Author Rebuttal · Authors · 2026-03-28
>
> 1. First, we thank the reviewer for identifying the strengths and originality of our paper.
>
> 2. It is true that our paper only addresses the case of Gaussian distributions. The main challenges with extending it to arbitrary probability distributions are:
>
>    - The 2-Wasserstein distance between two arbitrary probability distributions need not have a closed-form expression.
>
>    - The optimal transport map from a probability distribution $X_1$ to $X_2$ need not be affine. In our paper, the optimal transport map $T$ from $X_1 = \mathcal{N}(\mu_1, \Sigma_1)$ to $X_2 = \mathcal{N}(\mu_2, \Sigma_2)$ is affine:
>      $$T(x) = \mu_2 + A(x - \mu_1) $$
>
>    - Moreover, our paper characterized the separation between Gaussian distributions in $L^2$ distance which has a clean closed-form expression involving the means and covariance matrices. This need not exist for arbitrary probability distributions.
>
>    - The main theoretical tool we used in the paper is Banach's fixed point theorem to guarantee the existence of a unique fixed point in the neighborhood of a Gaussian distribution. The above three facts make proving one-step contraction in $W_2$ distance, which we did for the Gaussian case in the paper in Lemma 4 in the paper, quite challenging.
>
> 3. In the paper we made empirical comparisons with Dense Associative Memory which is a modern Hopfield network, because we thought that is an appropriate comparison. However, we realize the importance of transformer style memory mechanisms. Hence, in our revision, we will add a comparison to energy transformers.
>
> 4. Each iteration of Algorithm 1 costs $O(Nd^3)$, dominated by $N$ eigen-decompositions of $d\times d$ matrices for the transport maps, so scaling to larger $N$ or $d$ would benefit from diagonal covariance approximations (reducing cost to $O(Nd)$) and low-rank covariance structure. We will add a discussion of computational complexity and scalability to the revised version.

---

> > ### Author Rebuttal · Reviewer_fasU · 2026-04-03
> >
> > I thank the authors for their response and the interesting work. I remain positive about the work.

---

### Official Review · Reviewer_2cGq · 2026-03-13

**Soundness:** 3
**Presentation:** 3
**Significance:** 3
**Originality:** 3
**Overall Recommendation:** 5
**Confidence:** 3

**Summary:**

In this paper, the authors propose an extension of Euclidean Dense Associative Memories (Eu-DAMs) that enables the storage and retrieval of Gaussian distributions rather than their representations as deterministic vectors. This is achieved by equipping the space of probability distributions with the Bures-Wasserstein geometry. Within this framework, the authors define a novel log-sum-exp energy over stored distributions, along with a retrieval dynamics that aggregates optimal transport maps in a Gibbs-weighted manner.

Experimental validation on both synthetic and real-world datasets, including image datasets (CelebA and CIFAR-10) and text datasets (text8 and the NLI corpus), demonstrates that the proposed model, referred to as Bures-Wasserstein Dense Associative Memory (BW-DAM), outperforms the Eu-DAM and Pixel-DAM baselines. At the same time, it achieves accurate retrieval and exhibits robustness consistent with the theoretical analysis provided in the paper.

**Compliance With Llm Reviewing Policy:**

Affirmed.

**Final Justification:**

I am satisfied with the authors’ rebuttal. Therefore, I maintain my positive score.

**Key Questions For Authors:**

See the weaknesses above.

**Limitations:**

Yes.

**Strengths And Weaknesses:**

**Strengths**

* The paper provides strong theoretical analysis. The claims are supported by appropriate definitions and theorems, with proofs provided in the appendix.

* The experimental validation is satisfactory and uses a reasonably diverse setup. The obtained results are convincing and support the applicability of the proposed method.

* The source code is provided, which ensures the reproducibility of the experimental results.

* To the best of my knowledge, the proposed BW-DAM model offers a new perspective on distributional storage in generative AI.

**Weaknesses**

* Although the authors mention extending the approach to broader families of probability distributions as future work, the presented method currently applies only to Gaussian distributions.

* While the experimental study is supported by strong theoretical analysis, assumptions made in the theory are explicitly validated in practice.

* The theoretical exposition could benefit from a more intuitive and reader-friendly presentation alongside the formal analysis.

*Minor comments*

* Page 1, lines 25--26 (left column): "(Sohl-Dickstein et al., 2015; Ho et al., 2020) learn distributions" $\rightarrow$ "(Sohl-Dickstein et al., 2015; Ho et al., 2020), which learn distributions".

* Page 2, lines 83--84 (left column): "analysis. (see Section 4)" $\rightarrow$ "analysis (see Section 4)."

* Page 2, lines 70--71 (right column): the notation $\mathbb{S}^d_{++}$ should be defined.

* Placing the related work section in the appendix to save space is somewhat nonstandard.

---

> ### Author Rebuttal · Authors · 2026-03-28
>
> 1. First, we thank the reviewer for identifying the strengths and originality of our paper.
>
> 2. It is true that our paper only addresses the case of Gaussian distributions. The main challenges with extending it to arbitrary probability distributions are:
>
>    - The 2-Wasserstein distance between two arbitrary probability distributions need not have a closed-form expression.
>
>    - The optimal transport map from a probability distribution $X_1$ to $X_2$ need not be affine. In our paper, the optimal transport map $T$ from $X_1 = \mathcal{N}(\mu_1, \Sigma_1)$ to $X_2 = \mathcal{N}(\mu_2, \Sigma_2)$ is affine:
>      $$T(x) = \mu_2 + A(x - \mu_1) $$
>
>    - Moreover, our paper characterized the separation between Gaussian distributions in $L^2$ distance which has a clean closed-form expression involving the means and covariance matrices. This need not exist for arbitrary probability distributions.
>
>    - The main theoretical tool we used in the paper is Banach's fixed point theorem to guarantee the existence of a unique fixed point in the neighborhood of a Gaussian distribution. The above three facts make proving one-step contraction in $W_2$ distance, which we did for the Gaussian case in the paper in Lemma 4 in the paper, quite challenging.
>
> 3. We shall make the theoretical results in the Appendix more easier to parse, by adding more accessible comments that describe the lemmas, in the revision.
>
> 4. We shall address all the minor corrections (typographical errors, reference formatting, notation section clutter) you suggested in the revision.

---

> > ### Author Rebuttal · Reviewer_2cGq · 2026-04-03
> >
> > I am satisfied with the authors’ rebuttal. Therefore, I maintain my positive score.

---

### Decision · Program_Chairs · 2026-04-30

**Decision:**

Accept (regular)

**Comment:**

The paper extends dense associate memories to Gaussian distributions equipped with Wasserstein distance. Reviewers were unanimously positive about the paper and it is recommended for acceptance.